# Age-related divergence of circulating immune responses in patients with solid tumors treated with immune checkpoint inhibitors

Chester Kao[1,6], Soren Charmsaz [1,6], Hua-Ling Tsai [1,6], Khaled Aziz[2], Daniel H. Shu[1], Kabeer Munjal[1], Ervin Griffin[1], James M. Leatherman[1], Evan J. Lipson [1], Yasser Ged[1], Jeannie Hoffman-Censits[1], Howard L. Li [1], Elsa Hallab[1], Madelena Brancati[1], Mari Nakazawa [1], Stephanie Alden[1], Christopher Thoburn [1], Nicole E. Gross [1], Alexei G. Hernandez[1], Erin M. Coyne [1], Emma Kartalia[1], Marina Baretti[1], Elizabeth M. Jaffee [1,3], Sanjay Bansal[4], Laura Tang[4], G. Scott Chandler [5], Rajat Mohindra[5], Won Jin Ho [1,3] ✉, Mark Yarchoan [1,3] ✉ & Daniel J. Zabransky [1,3] ✉

Most new cancer diagnoses occur in patients over the age of 65. The composition and function of the immune system changes with age, but how the aged immune system affects responses to immune checkpoint inhibitor (ICI) cancer therapies remains incompletely understood. Here, using multiplex cytokine assay and high-parameter mass cytometry, we analyze prospectively collected blood samples from 104 cancer patients receiving ICIs. We find aged patients ($\geq 65$-years-old; $n = 54$) derive similar clinical outcomes as younger patients ($n = 50$). However, aged, compared to young, patients have divergent immune phenotypes at baseline that persist during ICI therapy, including diminished cytokine responses, reduced pools of naïve T cells with increased relative expression of immune checkpoint molecules, and more robust effector T cell expansion in responders compared to non-responders. Our study provides insights into age-stratified mechanisms of ICI effects while also implying the utility of age-tailored immunotherapeutic approaches.

Increases in human life expectancy have led to rapidly aging populations with higher incidence and prevalence of cancer. More than half of newly diagnosed cancers and 72% of all cancer deaths in the United States occur in people over 65 years of age[1]. Cancer incidence is 10-fold higher and cancer death rate is 16 times greater in people older than 65 years[2,3]. The clinical management of cancer in aged patients can be complicated by age-related reductions in normal organ function and co-morbid conditions, resulting in a higher risk of toxicity and complications with cytotoxic therapies such as chemotherapy[4,5]. As compared to other forms of cancer therapy, immune checkpoint inhibitors (ICIs) are often well-tolerated, and therefore may be an attractive treatment modality for older patients with cancer. Although elderly patients are underrepresented in prospective clinical trials of ICIs[6], numerous published studies and meta-analysis reproducibly

[1]Department of Oncology, Johns Hopkins University School of Medicine., Baltimore, MD, USA. [2]Department of Radiation Oncology and Molecular Radiation Sciences, Johns Hopkins University School of Medicine., Baltimore, MD, USA. [3]Convergence Institute, Johns Hopkins University., Baltimore, MD, USA. [4]Genentech Inc, South San Francisco., California, USA. [5]F. Hoffman-La Roche Ltd., Basel, Switzerland. [6]These authors contributed equally: Chester Kao, Soren Charmsaz, Hua-Ling Tsai. ✉e-mail: wjho@jhmi.edu; mark.yarchoan@jhmi.edu; dzabran1@jhmi.edu

demonstrate that patients aged 65 to 75 years respond similarly as patients under 65 years[7–11]. In some cancers such as melanoma, advanced age may be associated with better clinical outcomes with ICI therapy[12,13], though the potential mechanisms underlying such associations are not completely understood.

With aging, the immune system undergoes complex changes in both the innate and adaptive immune response, leading to a decreased capacity to respond to and eliminate pathogens as well as cancer cells[14]. The aging immune system is characterized by a number of changes in the function of immune cells and their responses to antigens. Immunosenesence, defined by an impaired ability to respond to new antigens coupled with persistent low-grade inflammation, the accumulation of memory T cells, and a decrease in naïve T cells can favor a lack of memory responses and has been associated with aging[15]. Additional changes in the adaptive immune system during aging include a loss of T cell receptor (TCR) diversity, impaired B cell function, and accumulation of immunosuppressive T regulatory (Treg) cells[16–18]. Aging is also characterized by dysregulated inflammation, termed "inflammaging", driven by defective organelle function, cellular stress, and activation of inflammasome signaling among other factors. This results in chronic basal inflammation and impaired ability to mount efficient innate and adaptive immune responses through production of interleukins such as IL-6, IL-1, TNF-α and immunomodulators including C-reactive protein (CRP)[19]. Finally, age-related changes in non-immune cells populations can also affect immune function. A hallmark of the normal aging process is the accumulation of senescent cells. Cellular senescence, a response characterized by growth arrest that limits the lifespan of cells and prevents uncontrolled proliferation, may serve to prevent malignant transformation[14]. Senescent cells can secrete pro-inflammatory and proteolytic factors including but not limited to IL-1, IL-5, IL-6, IL-8, IL-13, IL-15, IL-18, GM-CSF, CCL2, and tumor necrosis factor (TNF) as part of the senescence-associated secretory phenotype (SASP) which has been connected with dysfunctional immune responses against cancer[18,20–25].

While the aging immune system has been well characterized in patients without cancer, the effects of age on the host response dynamics to ICI treatment in patients with cancer is not fully understood. Understanding how ICIs reinvigorate immune surveillance and augment tumor-directed T cell responses in the remodeled immune system of aged patients remains a critical gap in knowledge that if addressed can identify additional opportunities for therapeutic intervention or improved ability to select individual patients for treatment with ICIs. Here, using prospectively collected, clinically annotated biospecimens from a pan-tumor cohort of patients receiving ICI therapy, we interrogate the circulating dynamics of the peripheral immune response to immunotherapy in young and aged individuals with cancer and show divergence in the composition and phenotypes of circulating immune cell populations and cytokine responses between young and aged patients treated with immune checkpoint inhibitors.

## Results

### Young and aged patients have similar clinical characteristics and outcomes with ICI treatment

From June 2021 to October 2022, we enrolled 124 patients who received ICIs as standard of care treatment for a variety of cancers at Johns Hopkins and prospectively followed them for at least 6 months in this observational study. Among these patients, 104 provided peripheral blood samples at baseline and approximately 1–5 months after initiation of a checkpoint inhibitor therapy and met criteria for inclusion for data analysis (Fig. 1A). The baseline demographics, tumor types, treatment regimens, and clinical outcomes of the patients in our cohort are shown in Table 1 and Supplementary Table 1. The median follow-up in this study cohort was 8.8 months from the baseline, as estimated using the reverse Kaplan-Meier method.

The median age at cancer diagnosis for the total cohort was 65 (range 20 to >90). The majority of patients (67.3%, $n = 70$) were male, and 28.8% ($n = 30$) of patients identified as Black. The most common tumor types were hepatocellular carcinoma (HCC) ($n = 30$, 28.8%) and renal cell carcinoma (RCC) ($n = 24$, 23.1%). Most patients ($n = 96$, 92.3%) were treated in the advanced/metastatic setting with 44.2% of patients ($n = 46$) having a history of prior oncologic systemic treatment, and 5.8% of patients ($n = 6$) having received prior ICI therapy before participating in the study. The average number of lines of prior systemic oncologic therapy in the cohort was 0.82. All 104 included patients received an anti-PD-1 or anti-PD-L1 ICI, and 22.1% ($n = 23$) received anti-PD-1 in combination with anti-CTLA-4.

To enable comparisons between patients by age, a cutoff of 65 years and older was selected to categorize patients as aged (51.9%, $n = 54$ of patients in this study). This cutoff was selected for several reasons. 65 years of age is a commonly used threshold to define older individuals in the U.S. and aligns with population cancer statistics which illustrate the median age of cancer diagnosis is around 65–67 years in the United States[26], allowing for interpretation of our results in the context of broader epidemiologic trends. In addition, recent evidence has suggested that age-related changes occur in a non-linear fashion at distinct chronological timepoints in the lifespan, with an age of 65 identified as a key point at which many of these changes occur[27], and that age related dynamics of RNA expression in circulating immune cells is distinct in healthy donors at age 65 and above compared to younger donors[28]. There were no significant differences in evaluated baseline demographic characteristics between aged (≥65 years) and young (< 65 years) patients (Table 1, $P > 0.05$).

When examining outcomes for patients with advanced/metastatic disease ($n = 96$), we found no significant difference for progression free survival (PFS) and overall survival (OS) by age group ($P = 0.1$ and $P = 0.46$, respectively; Fig. 1B, C). Among patients with advanced/metastatic and Response Evaluation Criteria in Solid Tumors (RECIST) 1.1 assessable disease ($n = 87$), aged patients achieved a response rate of 35.4% ($n = 17/48$) compared to 23.1% ($n = 9/39$) in younger patients (Fig. 1D, $P = 0.25$). Aged patients had a trend towards a higher rate of any grade immune-related adverse events (irAEs) with an overall rate of 55.6% ($n = 30/54$) compared to 38% in younger patients ($n = 19/50$) (Fig. 1E, $P = 0.08$). While an association between irAEs and age was not statistically significant in our cohort, prior studies have linked advancing age with increased risk of developing irAEs[29]. When assessing the time to onset of first irAE and incorporating follow up time, there was no significant difference between cumulative probability of irAEs based on patient age group (Fig. 1F, P = 0.12). As previous studies have suggested a connection between the development of irAEs and a higher likelihood of experiencing a response to ICI treatment (particularly with targeting of the PD-L1/PD-1 axis)[30], we examined this association in our young and aged patients. While this analysis is limited by sample size, we found that there was a non-significant trend for patients with irAEs to also respond to ICI treatment, with a stronger association observed in aged patients (76.5% of aged responders developed irAEs compared to 55.6% of young responders) (Supplementary Fig. 1).

There are currently three FDA-approved biomarkers for determining eligibility for ICI treatment: tumor mutational burden (TMB), PD-L1 expression, and high microsatellite instability (MSI-H) and mismatch repair deficient (dMMR) status[31]. We sought to evaluate whether there were age-related differences in these three biomarkers within our cohort. In total, there were 50 patients with available TMB data of which 48.0% ($n = 24$) were 65 years or older. There was no significant difference in TMB between aged (TMB = 6.6 mut/Mb) and young patients (TMB = 7.0 mut/Mb) (Fig. 1G, $P = 0.57$). Thirty-six patients had available MSI/dMMR status of which 58.3% ($n = 21$) were 65 years or older. There was no significant difference based on MSI/dMMR status of which 9.5% ($n = 2$) of aged and 13.3% ($n = 2$) of young patients having

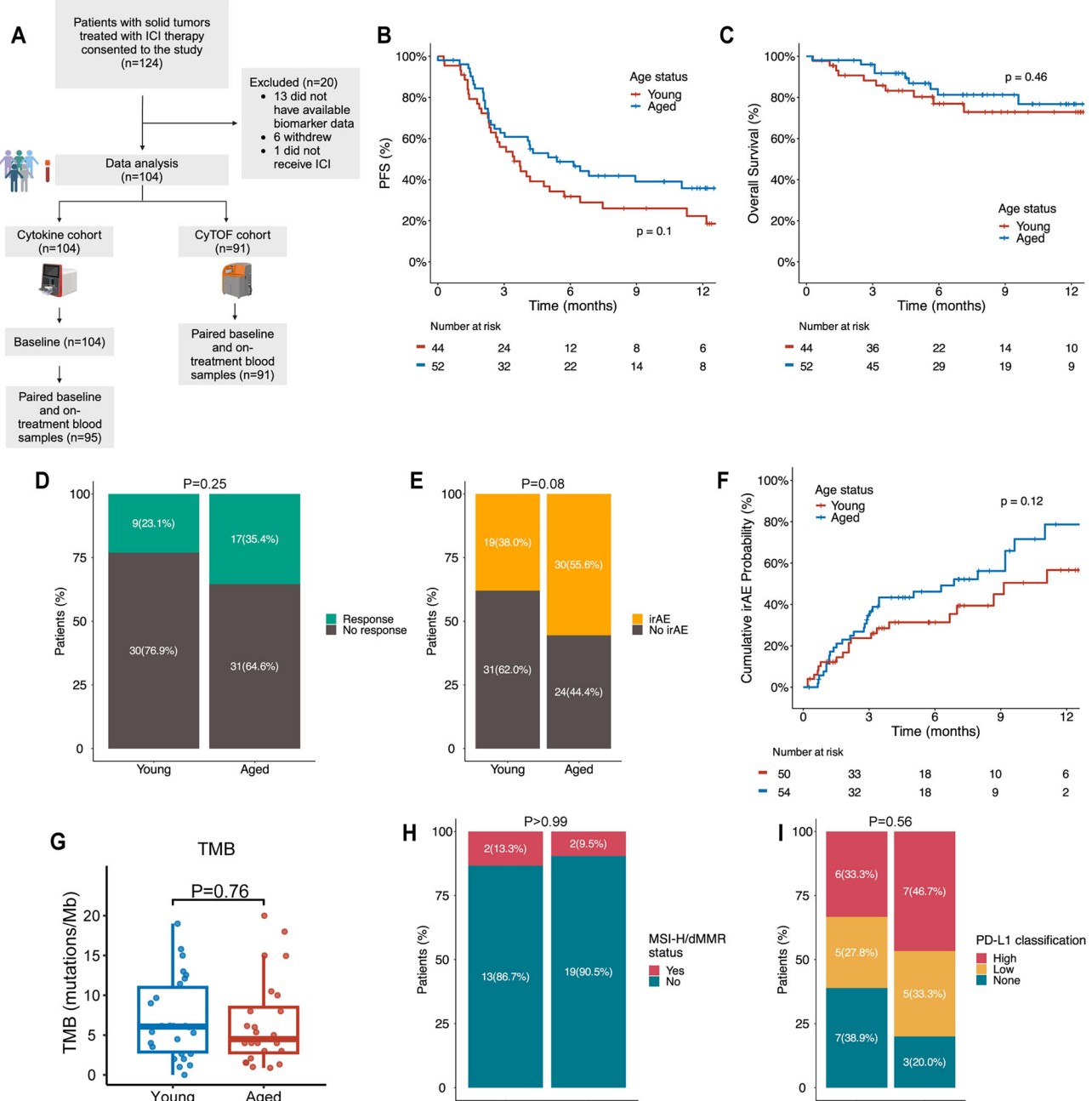

**Fig. 1 | Clinical outcomes, toxicity, and biomarker distribution in young and aged patients with solid malignancies. A** Consort diagram showing study enrollment and downstream data analysis. **B** Progression free survival (PFS) and (**C**) overall survival (OS) based on age group in patients with advanced/metastatic disease (*n* = 96). **D** Best objective response based on age group in patients with Response Evaluation Criteria in Solid Tumors (RECIST) assessable advanced/metastatic disease (*n* = 87). **E** Proportion of patients who developed immune-related adverse events (irAE), divided by age group. **F** Cumulative irAE for time to first irAE onset, incorporating follow up time, in the total cohort (*n* = 104). Distribution of three biomarkers: **G** Tumor mutation burden (TMB) in mutations/megabase (mut/Mb) (*n* = 50) according to age group. **H** Proportion of patients with

microsatellite instability high/mismatch repair deficient (MSI-H/dMMR) tumors in the young and aged cohorts (*n* = 36). **I** PD-L1 classification (*n* = 33) for patients with available biomarker data. In (**G**), the box and whisker plot shows the median, interquartile range (IQR), minimum/maximum values, and outliers. Time to event analyses were visualized with Kaplan-Meier curves for PFS and OS and reverse-Kaplan Meier curves for time to irAE onset, and statistical comparisons were performed utilizing two-sided log-ranked test. Statistical comparisons between quantitative measurements were performed using two-sided Wilcoxon rank-sum test and Fisher's exact test for categorical variables. Portions of this figure were created in BioRender. Leatherman, J. (2025) https://BioRender.com/x33d053. Source data are provided as a Source Data file.

MSI-H or dMMR status (Fig. 1H, *P* = 1.0). Lastly, 33 patients had available PD-L1 expression status of which 45.5% (*n* = 15) were 65 years or older. There was no significant difference in our study population based on PD-L1 expression status with 46.7% (*n* = 7) of aged patients and 33.3% (*n* = 6) of young patients having high PD-L1 (Fig. 1I, *P* = 0.56).

## Aged patients have diminished circulating cytokine changes in response to ICI treatment

As circulating cytokine levels have been found to impact ICI treatment responses, and ICIs themselves can modulate peripheral cytokine concentrations, we sought to determine if differences in baseline and

**Table 1 | Baseline characteristics of the cohort**

| Demographic Categories | All patients (n = 104) | Age ≥ 65 (n = 54) | Age < 65 (n = 50) | p-value |
|---|---|---|---|---|
| Age on study – yr | | | | |
| Median (range) | 65 (20 to > 90) | 71 (65 to > 90) | 55 (20–64) | |
| Sex – no. (%) | | | | 0.84 |
| Male | 70 (67.3) | 37 (86.5) | 33 (66.0) | |
| Female | 34 (32.7) | 17 (31.5) | 17 (34.0) | |
| Race – no. (%) | | | | 0.23 |
| White | 67 (64.4) | 39 (72.2) | 28 (56.0) | |
| Black | 30 (28.8) | 12 (22.2) | 18 (36.0) | |
| Other | 7 (6.8) | 3 (5.6) | 4 (8.0) | |
| Autoimmune history – no. (%) | | | | 1.00 |
| Yes | 13 (12.5) | 7 (13.0) | 6 (12.0) | |
| No | 91 (87.5) | 47 (87.0) | 44 (88.0) | |
| Cancer group – no. (%)[A] | | | | 0.13 |
| Gastrointestinal (GI) | 38 (36.5) | 22 (40.7) | 16 (32.0) | |
| Genitourinary (GU) | 34 (32.7) | 20 (37.0) | 14 (28.0) | |
| Upper aerodigestive (UAD) | 10 (9.6) | 2 (3.7) | 8 (16.0) | |
| Skin | 8 (7.7) | 5 (9.3) | 3 (6.0) | |
| Other | 14 (13.5) | 5 (9.3) | 9 (18.0) | |
| Disease stage – no. (%) | | | | 0.15 |
| Early | 8 (7.7) | 2 (3.7) | 6 (12.0) | |
| Advanced/Metastatic | 96 (92.3) | 52 (96.3) | 44 (88.0) | |
| Treatment regimen – no. (%) | | | | 0.94 |
| ICI monotherapy | 44 (42.3) | 23 (42.6) | 21 (42.0) | |
| Dual ICI combination therapy[B] | 23 (22.1) | 11 (20.4) | 12 (24.0) | |
| ICI with targeted therapy or chemotherapy | 37 (35.6) | 20 (37.0) | 17 (34.0) | |
| Prior oncologic systemic therapy – no. (%) | | | | 0.32 |
| Yes | 46 (44.2) | 21 (38.9) | 25 (50.0) | |
| No | 58 (55.8) | 33 (61.1) | 25 (50.0) | |
| Average lines of prior oncologic systemic therapy – no. | 0.82 | 0.59 | 1.06 | 0.11 |
| Prior ICI therapy – no. (%) | | | | 1.00 |
| Yes | 6 (5.8) | 3 (5.6) | 3 (6.0) | |
| No | 98 (94.2) | 51 (94.4) | 47 (94.0) | |
| irAE status – no. (%) | | | | 0.08 |
| Grade 1 or higher irAE | 49 (47.1) | 30 (55.6) | 19 (38.0) | |
| No irAE | 55 (52.9) | 24 (44.4) | 31 (62.0) | |
| Objective Response – no. (%)[C] | (n = 87) | (n = 48) | (n = 39) | 0.25 |
| Yes | 26 (29.9) | 17 (35.4) | 9 (23.1) | |
| No | 61 (70.1) | 31 (64.6) | 30 (76.9) | |

[A]Cancer groups are defined as: Gastrointestinal (GI) – Biliary tract, pancreatic, hepatocellular, colorectal, gastric, gastrointestinal neuroendocrine, and esophageal cancers. Genitourinary (GU) – Bladder, prostate, and renal cancers. Upper aerodigestive (UAD) – Head and neck and lung cancers. Skin – cutaneous squamous cell, melanoma, and Merkel cell cancers. Other – Adrenal, breast, cervical, endometrial, sarcomas, and vulvovaginal cancers. [B]All dual combination immune checkpoint inhibitor (ICI) therapy consisted of ipilimumab + nivolumab. [C]This group only included patients with evaluable disease by Response Evaluation Criteria in Solid Tumors (RECIST). Further details on specific cancer types, ICI used, immune related adverse event (irAE) grading, and response are detailed in Supplementary Table 1. Statistical comparisons between quantitative measurements were performed using two-sided Wilcoxon rank-sum test without adjustments for multiple comparisons. yr year, no. number.

post-treatment plasma cytokine levels exist in aged versus young patients using a high-throughput 32-plex cytokine assay in all 104 patients[32–34]. None of the measured cytokines were significantly different at baseline between patient age groups. When reviewing cytokines that have been associated with SASP, a phenotype which may be more common in cells in aged patients, there was a trend toward higher CXCL9 and IL-6 levels in aged compared to younger patients

(Fig. 2A, $P = 0.06$ and $P = 0.08$, respectively). While this overall lack of age-driven differences in baseline cytokine levels was initially surprising, comparisons of baseline circulating cytokines in our study population are likely distinct from previous studies of aging in healthy individuals where comparisons are typically made across a wider spectrum of the lifespan[35].

The presence of cancer is known to alter circulating cytokine levels in patients, even with localized disease, though the exact changes reported in the literature are heterogeneous and vary by study, tumor type, and other factors[36–38]. In light of this, as well as due to the diversity of tumor types in our cohort, we investigated whether baseline cytokine levels were more strongly influenced by a patient's cancer type. Among the 4 most common tumor types, HCC, RCC, bladder cancer (BC), and head and neck cancer (HNC) in our cohort, baseline IL-8, IL-13, IL-17f, CCL5, CXCL9, CXCL10, VEGF-α, and GM-CSF differed between two of the tumor types, and IL-8, IL-17f, and CXCL10 differed significantly among the four tumor types (Supplementary Fig. 2A, B, $P < 0.05$). However, IL-6 and CXCL9, baseline concentrations did not significantly differ by patient age group when evaluating within the same tumor type (Supplementary Fig. 2C, $P > 0.05$). These data suggest that the baseline, i.e., pre-ICI, cytokine environment is not as strongly influenced by the patient's age compared to the tumor type, underscoring the need to investigate the dynamic changes during ICI therapy and how they relate to patient age.

Among the 104 patients, 95 patients had available on-treatment samples to assess the cytokine changes that occur after ICI exposure. 83.2% ($n = 79$) of patients had on treatment samples collected within 2 months of starting ICI therapy, 13.7% ($n = 13$) collected after 2 months but within 3 months, and only 3.2% ($n = 3$) of patients having samples collected 3-5 months post-initiation of ICI therapy. For visualization purposes, scaled concentrations for all 32 cytokines at baseline ($n = 104$) and on treatment ($n = 95$) are shown in Supplementary Fig. 3A-B and fold change on treatment ($n = 95$) in Fig. 2B. On treatment, aged patients had lower absolute concentrations of CCL2, CCL5, IL-18, and G-CSF (Supplementary Fig. 3C, $P < 0.05$) compared to younger patients. Notably, IL-18 been reported to act synergistically with other pro-inflammatory cytokines to promote antitumor immune responses and potentiate the effects of ICIs, and lower levels of G-CSF are associated with a less immunosuppressive tumor immune microenvironment[39–41].

We next sought to determine if patient age affected the circulating cytokine responses to ICIs. As patients had available baseline and on treatment plasma samples, we calculated fold change values for each measured cytokine in each patient. Aged patients had a lower fold change in IL-10, IL-12(p40), IL-15, IL-18, and CCL2 after ICI exposure (Fig. 2C, $P < 0.05$). These changes appeared specific for individual cytokines rather than broader classes of cytokines associated with specific functional programs (e.g. Th1, Th2, Th17). When cytokines were grouped into categories as denoted in Fig. 2B we did not observe any significant changes in cytokines by category in young vs. aged patients aside from "Treg" which consisted solely of IL-10 (Supplementary Fig. 4). These results suggest a differential secretory capacity based on age with aged patients having diminished circulating cytokine responses to ICI treatment. Functionally, these changes in aged patients may have both positive and negative effects for ICI efficacy. For instance, decreased IL-10 and CCL2 have been associated with improved ICI outcomes[42,43] but a lack of IL-18 may hamper memory CD8 + T cell formation post-ICI treatment[44].

As we did identify differences in cytokine expression based on tumor type (Supplementary Fig. 2), we next evaluated if there were any age-related differences in cytokine expression dynamics in the subset of patients with HCC or RCC, the two most common tumor types in our cohort (Supplementary Fig. 5 and Supplementary Fig. 6). The results from these analyses were overall consistent with those from the full cohort in terms of trends in fold change values (on treatment/baseline)

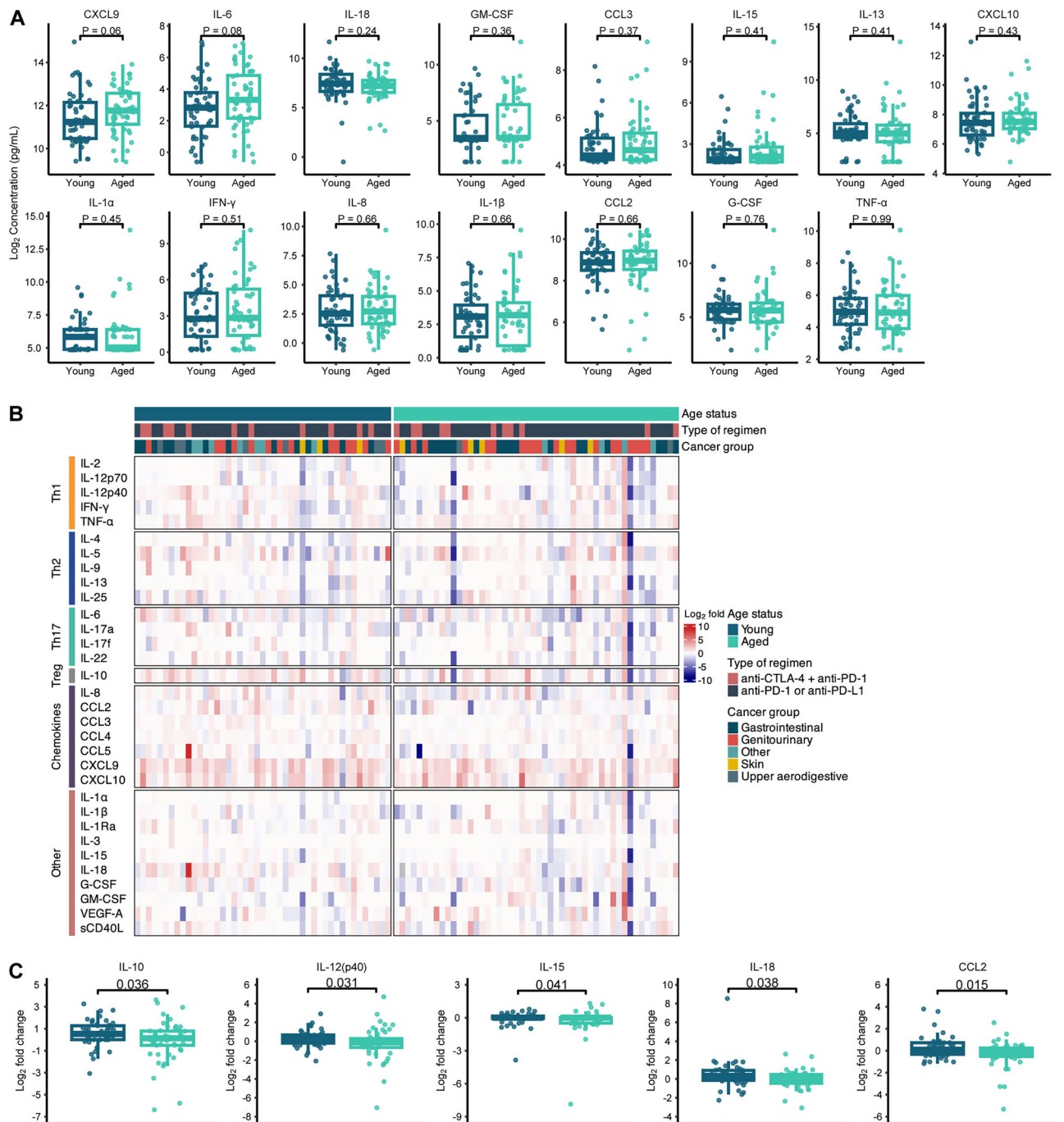

**Fig. 2 | Baseline and cytokine responses in aged patients after ICI exposure is unique from young patients. A** Senescence associated secretory phenotype (SASP)-related cytokines at baseline (aged, $n = 54$; young, $n = 50$). **B** Heatmap visualization of log$_2$ transformed fold change of 32 cytokines for patients stratified by age group after start of immune checkpoint inhibitor (ICI) treatment (aged, $n = 50$; young, $n = 45$). **C** Cytokines with statistically different levels between age

groups for log$_2$ transformed fold change after ICI treatment (aged, $n = 50$; young, $n = 45$). In (**A**, **C**), the box and whisker plots show the median, interquartile range (IQR), minimum/maximum values, and additional marking of outliers. Statistical comparisons between quantitative measurements were performed using a two-sided Wilcoxon rank-sum test without adjustment for multiple comparisons. Source data are provided as a Source Data file.

among the measured cytokines. There were, however, cytokine dynamics after ICI exposure unique to HCC, namely a significantly lower fold change in G-CSF in older patients. Interestingly, G-CSF production in some cancers is associated with higher PD-L1 expression and other features of immune exhaustion[45,46].

While we did not observe any clear confounding factors based on the characterization of our young and aged cohorts (Table 1), there were some demographics that trended differently. For example, young

patients were more likely to receive more lines of prior therapy, with an average of 1.06 prior lines compared to 0.59 in aged patients, though this was not statistically significant ($P = 0.11$). Therefore, we next incorporated multivariable models, adjusting for cancer type (GU vs. GI, all others vs. GI) and if patients had received prior oncologic systemic therapy as these factors may influence cytokine milieus (among other biological factors) into our analysis[47,48]. We observed similar trends in the fold change of cytokines as we had identified in our initial

analysis (such as decreased IL-10, IL-12(p40), IL-15, IL-18, and CCL2 expression in aged patients), though only reduced fold change in CCL2 and IL-18 in aged patients remained as a statistically significant findings (Supplementary Fig. 7). Interestingly, multivariable analysis highlighted new cytokines that have lower changes in expression with ICI treatment in aged patients, including IL-6, CCL5, IL-3, and G-CSF.

In sensitivity multivariable analyses examining circulating cytokine levels and responses to ICI treatments in relation to continuous age per decade, similar trends in cytokine fold change were observed. A notable example is the significantly diminished fold change in CCL2 expression with ICI treatment in older patients being identified (Supplementary Fig. 8), which supports the robustness of results from the initial analysis in which patients were grouped based on an age cutoff of 65.

Taken together, we observed a stronger difference in baseline cytokine expression based on tumor type rather than age group in patients with cancer but showed that advanced age is associated with a decreased magnitude of peripheral cytokine changes after ICI initiation. This is suggestive of differences in the biological effects of ICI treatments across age groups that may have functional consequences in an anti-tumor immune response warranting further investigation into circulating immune cell populations that may also be differentially impacted by ICIs depending on patient age.

### Cytokine dynamics differ in ICI responders depending on patient age

Next, we evaluated the impact of age on cytokine dynamics in relation to ICI response in our cohort. Multivariable models revealed several significant differences in cytokine expression between responders and non-responders within age cohorts (Supplementary Fig. 9). There were differences in the baseline and on-treatment levels of multiple cytokines (IL-2, IL-12p70, IL-17a, CCL2, IL-1α, VEGF-α, sCD40L), with significant interaction effects between age and response status across these cytokines. While aged patients had smaller increases in CCL2 expression after ICI treatment, interestingly, an increased fold change in CCL2 levels predicted non-response in young patients but response in aged patients (adjusted interaction $P = 0.02$). As previously mentioned, reduced levels of CCL2 have been associated with improved ICI outcomes, but our data highlights an opportunity to better understand the impact of peripheral CCL2 level on ICI responses in the context of older patients in future studies.

### Aged patients have a unique baseline circulating immune composition and altered immune cell phenotype dynamics associated with response to ICI treatment

We identified 91 patients that had paired pre- and on-treatment peripheral blood mononuclear cells (PBMCs) collected (characteristics in Supplementary Table 2). We found no significant differences in total white blood cells and a standard differential of neutrophils, immature granulocytes, lymphocytes, monocytes, and eosinophils at baseline (Supplementary Table 3, $P > 0.05$) based on age group. To better characterize the circulating immune cell populations in our patient cohort, we performed Cytometry by Time-of-Flight (CyTOF) on pre- and on-treatment PBMCs using a 37-antibody panel (Supplementary Data 1). This CyTOF panel allowed assessment of proportions of natural killer (NK) cells (CD56), type-1, type-2, vs. type-17 CD4 T cell phenotypes (CXCR3, CCR4, CCR5, CCR6, GATA3, RORγ) and Tregs (FOXP3), along with abundance of naïve compared to antigen-experienced T cell states (CD45RA, CCR7, CD45RO), and the promotion of T cell activation/exhaustion (GZMB, KI67, PD-1, 4-1BB, LAG-3) (Supplementary Data 2). From the expression profiles of the annotated cluster heatmap and accompanying UMAP visualization of unsupervised clustering results, we identified 15 unique immune cell groups that included previously defined subsets such as CD3$^+$CD4$^+$ helper T cells (Th), CD3$^+$CD8$^+$ cytotoxic effector T cells (TcEFF), CD3$^+$CD4$^-$CD8$^-$ double negative T cells (DNT), and NK cells (Fig. 3A, B)

and measured them as a proportion of the total immune cells analyzed.

At baseline, proportions of cytotoxic T cell populations including TcEFF, cytotoxic effector memory CD8+ (TcEM) T cells, and cytotoxic central memory CD8+ (TcCM) T cells were similar between aged ($n = 49$) and young ($n = 42$) patients (Fig. 3C, Supplementary Fig. 10A, $P > 0.05$). Aged patients had a significantly lower proportion of B cells, and naïve CD8+ (TcN (CD8$^+$CD45RA$^+$CCR7$^+$)) and naïve CD4+ helper (ThN cells (CD4$^+$CD45RA$^+$CCR7$^+$)) T cells but higher proportion of NK cells compared to young patients (Fig. 3C, $P < 0.05$). After initiation of ICI therapy, TcN and ThN cells were less frequent in aged patients (Fig. 3D, Supplementary Fig. 10B). Further, these differences in naïve TcN and ThN cell abundances held true when evaluating them as a proportion of total cytotoxic T or helper T cells respectively (Supplementary Fig. 11A, B) with a lower fold change in the proportion of TcN out of total Tc cells in aged compared to young patients (Supplementary Fig. 11C). Tumor type appeared to not be a major driver of this difference in TcN cells, as aged patients had a lower proportion of TcN cells when considering those with HCC and RCC alone (the two most common tumor types in our cohort) as well as when evaluating all other tumor types with HCC and RCC excluded (Supplementary Fig. 12A, B).

To further characterize these 15 immune groups, we performed further detailed annotation clustering to identify 25 unique immune populations including different subtypes of TcEFF, NK, and DNT cells (Supplementary Fig. 13A). Within the immune cell groups that differed by prevalence in young versus aged patients, we identified unique subclusters based on the expression of surface/functional markers that were found at lower abundance in aged compared to young patients including B_I (CCR6$^+$CCR7$^+$) and DNT_II (CD45RA$^+$GZMB$^-$) with a higher proportion of NK_II (CD16$^{hi}$GZMB$^+$NKG2D$^+$) cells (Supplementary Fig. 13B, $P < 0.05$). The lower abundance of CCR7$^+$ B cells in the B_I cluster, may suggest that aged patients have fewer B cells primed to traffic to the lymph node or into tumors[49]. As tumor infiltrating B cells have been implicated in the response to ICI[50], this highlights a potential difference in the overall immune response to ICIs in aged compared to young patients. These observations suggest that aged patients have lower proportion of naïve T cells, functionally mature B cells capable of responding to antigen challenge, and "naïve" antigen responsive DNT cells while a higher proportion of activated NK cells. After initiation of ICI treatment, this observation stayed consistent (Fig. 3D, $P < 0.05$) and with the addition of two more unique immune cell clusters: lower DNT_I (CD27$^+$CD28$^+$TIGIT$^+$Ki67$^+$) and higher NK_I (GZMB$^+$CCR3$^+$CCR4$^+$CXCR3$^+$Ki67$^+$) cells in aged patients (Supplementary Fig. 13C, $P < 0.05$). A lower proportion of DNT cells expressing TIGIT and Ki67 suggests that aged patients have less "activated" DNTs after ICI exposure compared to younger patients, but they have a higher proportion of proliferating Ki67+ NK cells, supportive of a dysregulated balance for the innate compartments over the adaptive in aged patients even after ICI treatment. In addition, aged patients had a significantly lower fold change in regulatory T (Treg) cells compared to young patients after ICI treatment (Supplementary Fig. 13D, $P < 0.05$). This may imply that aged patients are less likely to have increased Tregs after ICI exposure and a potential subsequent limiting of inhibitory effects on anti-tumor activity mediated by Tregs.

Multivariable analyses were conducted to examine the adjusted effect of age on circulating immune cell populations in our cohort. Similar to the multivariable analysis of the cytokine data, trends in cellular abundances across the 15 unique cell type groups compared young and aged cohorts remained robust, regardless of whether confounding adjustments were applied to the analysis (Supplementary Fig. 14). This included significantly lower naïve cytotoxic and helper T cells both at baseline and on treatment as well as higher proportions of NK cells on treatment in aged patients. In addition, we again observed a decreased fold change in Treg proportions in aged patients

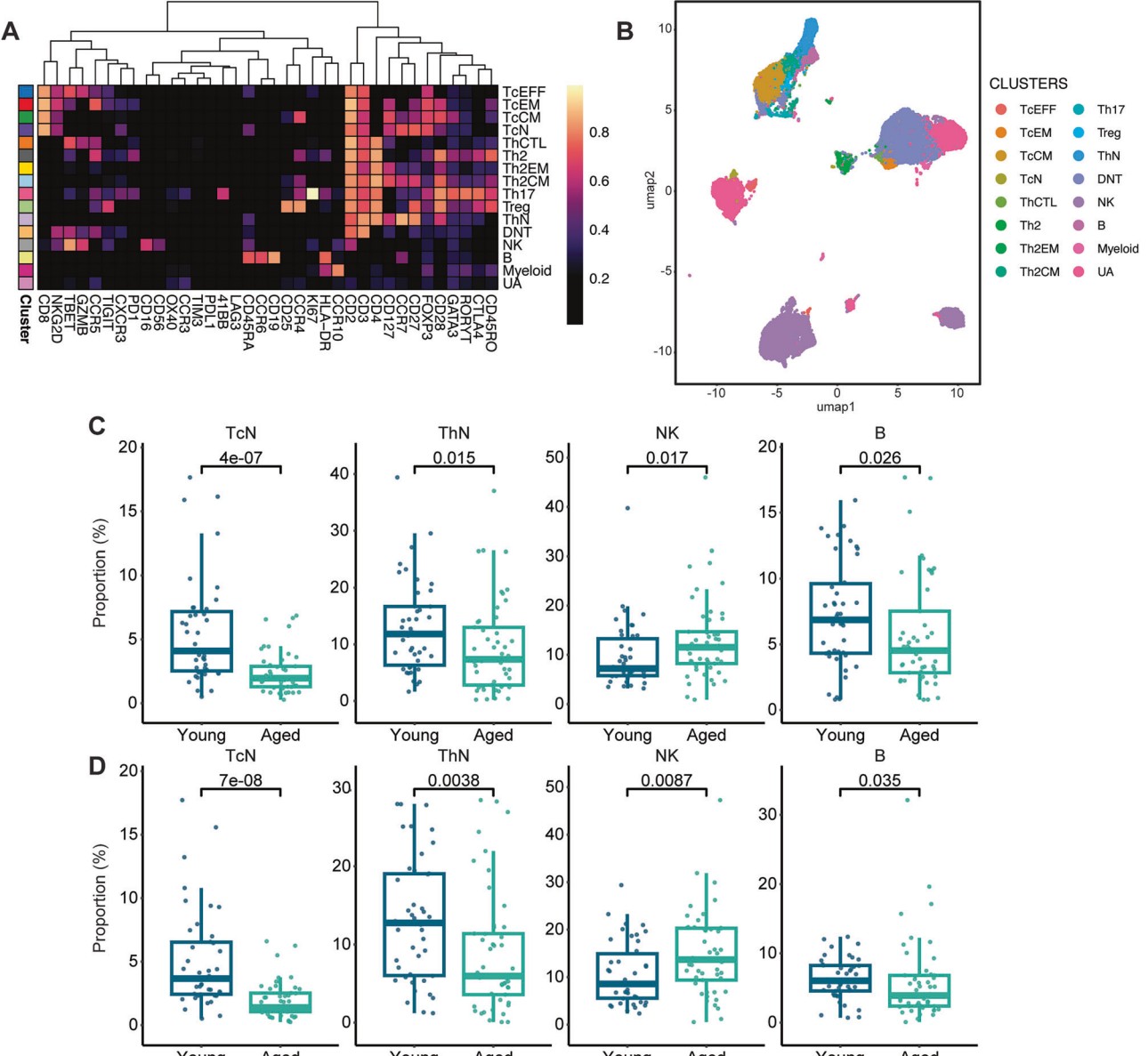

**Fig. 3 | Aged patients have a diminished pool of naïve T and B cells and higher NK cells pre- and post-treatment compared to young patients. A** Heatmap showing scaled expression of each cluster identified after CyTOF was performed on pre and post-treatment peripheral blood mononuclear cell (PBMC) samples with a 37-marker panel. A FlowSOM algorithm was used to generate metaclusters which were annotated into a final 15 clusters (immune groups). Scaled expression profile for each cluster is shown. **B** UMAP plots visualizing the annotated clusters (500 cells per sample). Proportion of the total cells selected major immune groups at (**C**) baseline and (**D**) on treatment (total $n = 91$, aged $n = 49$, young $n = 42$). In (**C, D**), the box and whisker plots show the median, interquartile range (IQR), minimum/maximum values, and additional marking of outliers. Statistical comparisons between quantitative measurements were performed using a two-sided Wilcoxon rank-sum test without adjustment for multiple comparisons. Source data are provided as a Source Data file.

after ICI treatment. Similar results were obtained when age was treated as a continuous variable (by decades) in multivariable analysis (Supplementary Fig. 15). Together, the results of this multivariable analysis suggest that major differences in the circulating immune repertoire in aged patients with cancer and receiving ICI treatment involve naïve T cell populations and the dynamic responses of regulatory T cells even when controlling for additional baseline characteristics in our patient population.

Next, we investigated age-related differences between the identified immune clusters and objective response to ICI treatment. In total, 78 patients met RECIST criteria for response assessment. Forty-six of these patients were ≥65 years old of which 17 were classified as responders and 29 as non-responders (ORR = 37.0%); 32 patients were <65 years of which 8 were responders and 24 non-responders

(ORR = 25%). Within aged patients, responders had a lower proportion of TcEFF and cytotoxic T helper (ThCTL) cells at baseline (Fig. 4A, $P < 0.01$), while younger responders had higher NK cells at baseline compared to young non-responders (Fig. 4A, $P < 0.05$). After treatment, aged responders continued to have lower peripheral TcEFF and ThCTL cells but increased central memory Th2 (Th2CM) compared to aged non-responders (Fig. 4B, $P < 0.01$), and younger responders had higher NK but lower effector memory Th2 (Fig. 4B, $P < 0.05$). Though aged responders had lower proportions of multiple T cell clusters both at baseline and on-treatment, they had a higher fold change of various T cell compartments including TcCM, TcEFF, TcEM, and Th2CM after ICI initiation accompanied by a lower fold change of myeloid cells compared to aged non-responders (Fig. 4C, $P < 0.05$). There were no observed significant differences in fold change of cell populations

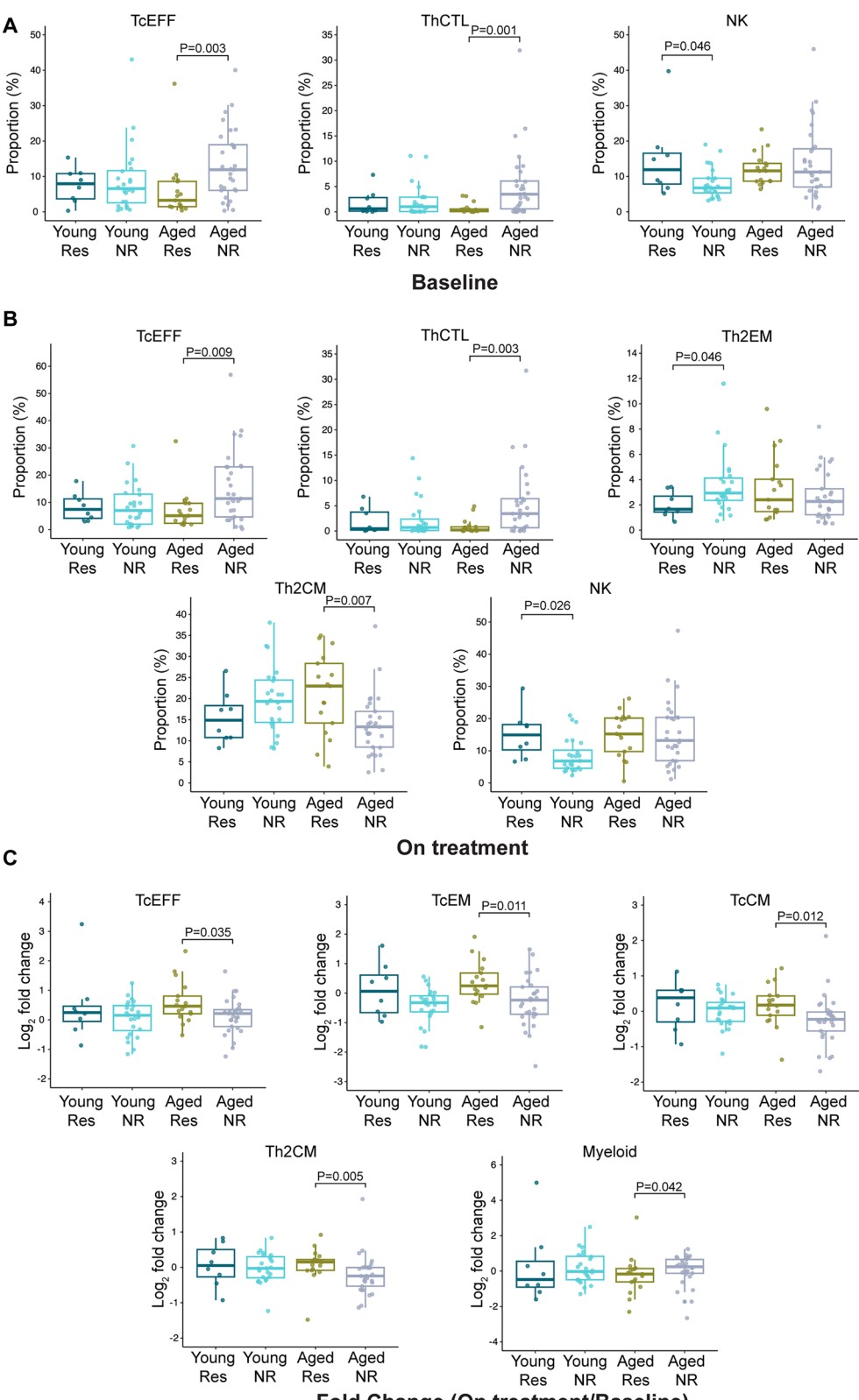

**Fig. 4 | Aged patients have a lower effective pool of adaptive immune compartments but an increase in effector/memory T cell compartments in immune checkpoint inhibitor responders.** The significantly different proportions of immune clusters between responders (Res) and non-responders (NR) within each age group at (**A**) baseline (**B**) on treatment. (**C**) Log$_2$ transformed fold change of immune cluster proportion between responders and non-responders within each age group. In total, $n = 78$ patients had advanced/metastatic and

RECIST assessable disease which consisted of aged responders ($n = 17$), aged non-responders ($n = 29$), young responders ($n = 8$), and young non-responders ($n = 24$). Box and whisker plots show the median, interquartile range (IQR), minimum/maximum values, and additional marking of outliers. Statistical comparisons between quantitative measurements were performed using a two-sided Wilcoxon rank-sum test without adjustment for multiple comparisons. Source data are provided as a Source Data file.

after ICI initiation for young responders versus non-responders. These patterns were also observed when we used multivariable linear regression models to analyze data, but no significant interaction effects between age and immune cell clusters that differed by age group were identified (Supplementary Fig. 16).

Overall, we observed, both at baseline and after ICI exposure, a relative abundance of activated proliferating NK clusters and lower pool of naïve T, B, and DNT cells expressing markers of antigen responsivity in aged patients compared to young, but paradoxically, a more dramatic increase in various effector/memory T cell compartments after ICI treatment in aged responders over non-responders. Taken together, even though aged patients have a lower effective pool of various adaptive immune compartments, T cells in aged patients that respond to ICI treatment appear to be particularly sensitive to immune checkpoint inhibition leading to an enhanced remodeling of T cell phenotypes in this group.

## Naïve T cells in aged patients have increased expression of immune checkpoints compared to naïve T cells from young patients

We next hypothesized that aged patients, despite possessing a lower pool of various T cell populations, may have immune cells that express distinct levels of markers that are associated with increased sensitivity to ICI. To this end, we assessed for differential immune marker expression between aged and young patients within each immune cluster by comparing the mean metal intensity (MMI) of unique marker-antibody conjugates. Due to the mechanisms and known functional consequences of immune checkpoint inhibitor therapies, we were particularly interested in markers of effector function/proliferation (granzyme B (GZMB), Ki67, TBET), memory (CCR7, CD45RO, CD45RA), and exhaustion (CTLA-4, LAG-3, PD-1, PD-L1, TIGIT, TIM3). We scaled the average MMI from the CyTOF data between young and aged patients within each cluster for individual markers to highlight the most age-related divergent markers. Scaled MMIs were calculated for each patient by normalizing to the mean and standard deviation of the individual marker for that immune cluster. When evaluating expression of effector and exhaustion/memory markers in each cluster, we noted multiple differences in either baseline or on treatment expression of proteins including granzyme B (GZMB), TBET, PD-1, PD-L1, TIGIT, and TIM3 across multiple immune clusters including B, TcEFF, DNT, TcEM, Th2, and Tregs (Supplementary Fig. 17A–F). Typically, immune cells from aged patients had higher expression of these markers with the exception of significantly increased PD-1 expression on TcEFF cells in young patients at baseline which did not persist after treatment with anti-PD-1. There were no significant differences in expression of the specified exhaustion and effector function markers in myeloid, NK, and additional Th cell subsets (Supplementary Fig. 18).

In humans, thymic involution leads to a reduction of the naïve T cell pool and maintenance is primarily dependent on peripheral division of existing clones rather than production of new ones[51–53]. In addition to the contraction of the pool of available naïve T cells with aging, there is a shift towards memory T cells or senescent cells[18,54–59]. Since generation of effector T cells is dependent on the naïve T cells available, phenotypic differences between naïve T cells by patient age group may explain the comparable T cell response after ICI initiation in aged patients despite a limited pool of available naïve T cells. For this analysis, we again normalized marker expression within the individual naïve cell cluster (TcN or ThN) and then compared expression between cells from young or aged patients. Aged patients had significantly higher expression of CD45RO and lower CD45RA and CCR7 in TcN cells and higher expression of CD45RO and lower CD45RA in ThN cells at baseline. While these cells are categorized as naïve and antigen inexperienced based on expression of markers including CD45RA, naïve cells from aged patients have a pattern that is reminiscent of a more memory-like phenotype in these otherwise naïve cells, albeit it at much lower expression of

markers than truly antigen experienced T cells express (Fig. 5A, B, $P < 0.05$). Furthermore, TcN had higher expression of canonical activation/exhaustion markers including PD-1, PD-L1, and TIGIT and of the transcription factor T-bet at baseline (Fig. 5C, D, $P < 0.05$), though again at lower levels in absolute terms compared to non-naïve T cell clusters. After ICI initiation, these differences are preserved with higher CD45RO and lower CD45RA and CCR7 expression in TcN cells and higher expression of CD45RO and lower CD45RA in ThN cells from aged patients (Fig. 5A, B, $P < 0.01$). There was also evidence of higher levels of TIGIT, and TIM3 but lower levels of LAG-3 in TcN cells of aged patients (Fig. 5C, D, $P < 0.05$). T-bet continued to be more highly expressed in TcN of aged patients (Fig. 5C, D, $P = 0.001$). This may suggest that in the immune cells of younger patients, the addition of anti-LAG-3 immunotherapies, thought to decrease tolerance to anti-PD-1 therapies, may be more effective than in aged patients, as immune checkpoint inhibition during activation of naïve cells can promote more robust effector T cell responses[60,61]. There were no differences PD-L1, TIM3, TIGIT, LAG-3, CTLA-4, or T-bet expression for ThN at baseline and on-treatment (Fig. 5A).

We then examined if expression of these immune checkpoints and other functional markers differed in naïve T cell subsets depending on patient ICI response in addition to age group. Within aged patients, we found that there were significant differences in functional marker expression between responders and non-responders to ICI therapy at baseline in the TcN, but not the ThN cluster. These included lower CCR5, T-bet, and Ox40 expression as well as higher CD27 expression in aged responders (Supplementary Fig. 19A). Aged responder TcN cells, on average, appeared more "naïve" (trend toward lower PD-1 and significantly lower TBET)[62,63], and have features that suggest potential to become activated/memory with the proper contextual signals (high CD27)[64,65], perhaps suggesting a higher likelihood of participating in the antitumor immune response with ICI treatment. When evaluating young responders vs. non-responders, the only differences observed in functional marker expression in naïve cell populations was significantly lower PD-1 in the TcN cluster in responders (Supplementary Fig. 19B). While the trend toward lower expression of PD-1 in circulating TcN cells at baseline was associated with ICI response in both age groups, the absolute expression levels of PD-1 were still higher in aged responder TcN cells compared to young responder TcN cells.

Together, these data show that conventionally identified naïve T cells are distinct in aged compared to younger patients, expressing relatively lower amounts of markers traditionally associated with a naïve state and relatively higher amounts of markers associated with a memory-like state. These data show that ICI treatment in aged patients results in a sustained expression of these markers, some of which are targets of currently used cancer immunotherapies, in these naïve T cells, highlighting unique biology that may impact the anticancer immune response after immunotherapy treatments.

## Integrated patient immune profiles highlight age-related differences driven by naïve T cell populations

Our analyses generated data related to cytokine expression and/or proportions of circulating immune cell subtypes from individual patients across the study cohort. We sought to understand if integrating these data points on a per patient basis across multiple time points could highlight trends that were associated with patient age or ICI response. We analyzed the low-dimensional representation of immune profiles by integrating measured cytokine levels and immune cell type abundances and performing Principal Component Analysis (PCA). Differences in average component scores were evaluated to identify the two principal axes that best facilitated comparisons based on patient age or ICI response.

At baseline, we found that young and aged patients were separated by components with relatively high contributions (PC5 and PC7, Fig. 6A) whereas there was not distinct separation based on patient ICI

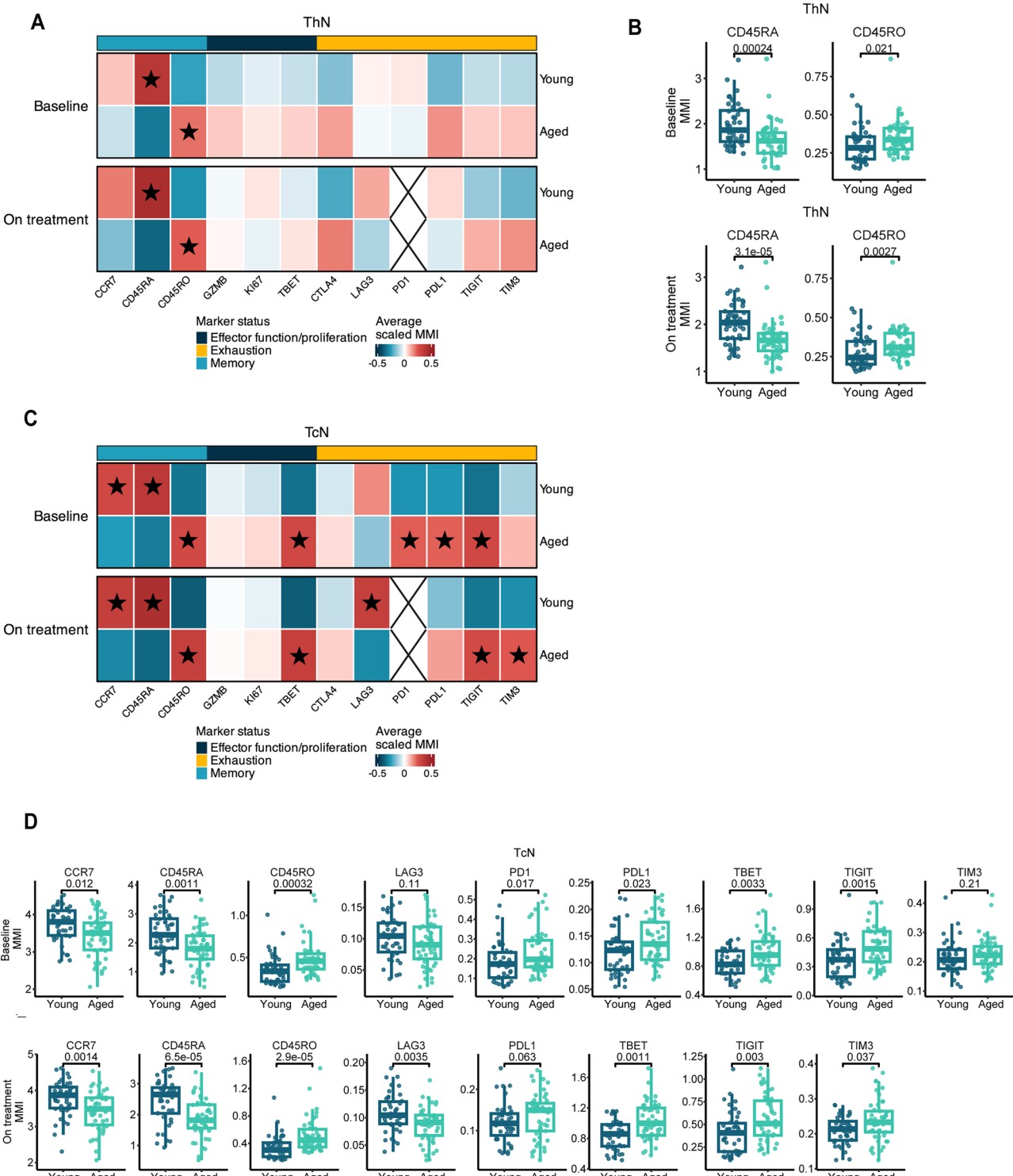

**Fig. 5 | Naïve T cells in aged patients have a unique phenotype compared to young patients before and after ICI treatment.** Heatmaps were generated for the average scaled mean metal intensity (MMI) between aged (*n* = 49) and young patients (*n* = 42) for 12 markers of interest related to T cell memory, effector/proliferation function, and exhaustion for 15 unique immune clusters at baseline and on treatment. Scaled MMIs were calculated for each individual marker within each immune cluster as defined from the annotation clustering from Fig. 3A. Scaling was performed for visualization purposes to highlight the most divergent markers by age group, and formal statistical comparisons were performed with a two-sided Wilcoxon rank-sum test on non-scaled MMIs without adjustment for multiple comparisons, with statistically significant comparisons (*P* < 0.05) indicated on the heatmaps with a star in the box of the more highly expressed marker along with absolute MMIs on box and whisker plots showing the median, interquartile range (IQR), minimum/maximum values, and additional marking of outliers. Results for ThN (**A**, **B**) and TcN (**C**, **D**) are presented. Analysis of PD-1 (PD1) expression in post-treatment samples was not included due to use of a competitive antibody as described in the methods. represents naïve immune clusters as labeled. Source data are provided as a Source Data file.

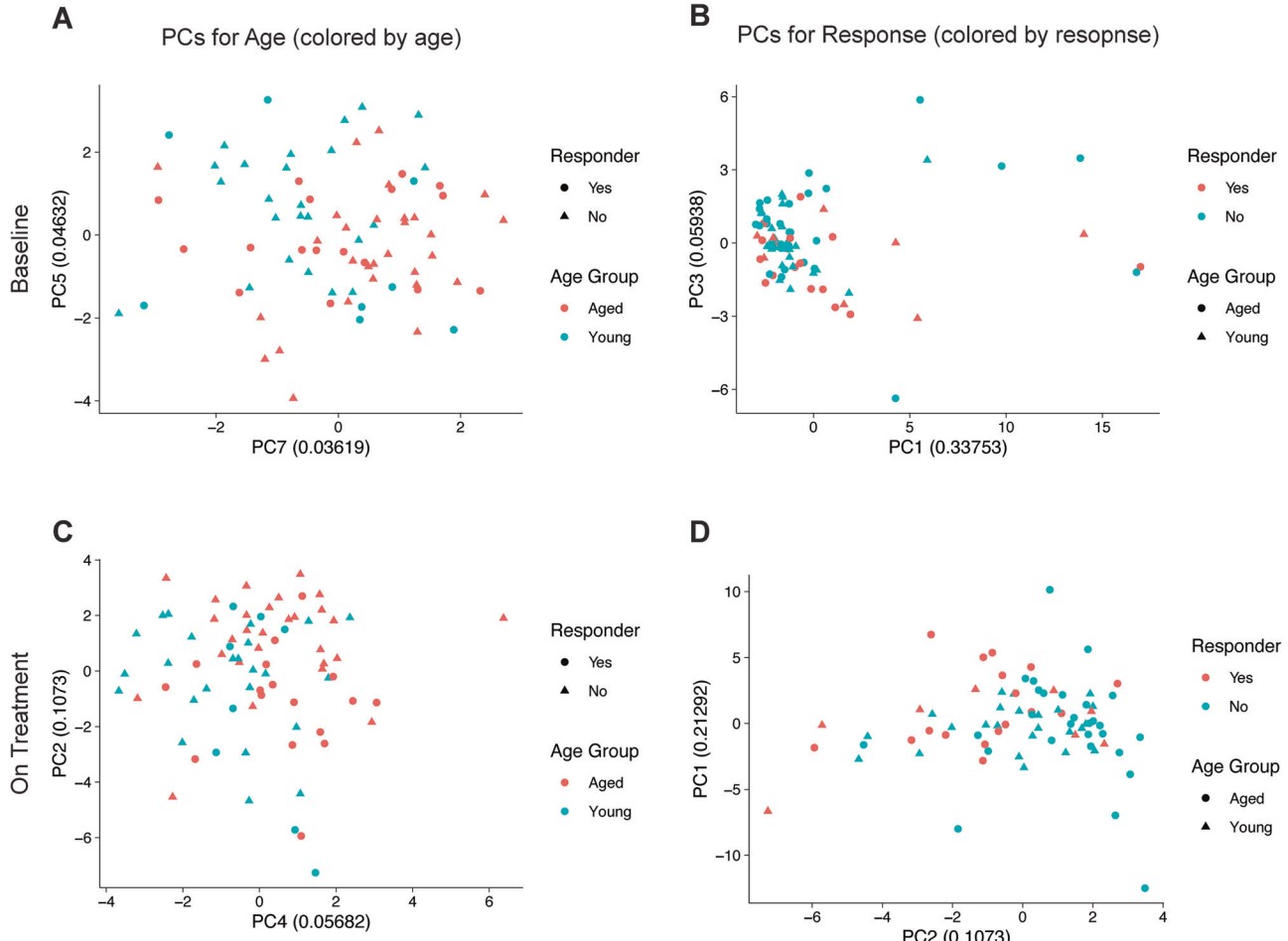

**Fig. 6 | Analysis of integrated cytokine and cellular immune profiles in young and aged patients treated with ICI therapy.** Principal component (PC) analysis of low-dimensional representations of immune profiles performed by integrating available cytokine and immune cell type abundance data on a per patient basis across the clinical cohort (*n* = 17 aged responders, 29 aged non-responders, 8 young responders, 24 young non-responders for (**A**, **B**) and *n* = 17 aged responders, 28 aged non-responders, 8 young responders, and 24 young non-responders for

(**C**, **D**). Patients without annotated clinical responses were not included in this analysis. Differences in average component scores were evaluated to identify the two principal axes that best facilitated comparisons based on age or response. Plots are presented baseline (**A**, **B**) and first on treatment (**C**, **D**) and for PCs for either age (left side) or immune checkpoint inhibitor (ICI) response status (right side). Source data are provided as a Source Data file.

response (Fig. 6B). This result is consistent with our prior analyses because principal component (PC)5 is directly correlated with naïve T cell proportions while PC7 is inversely correlated with naïve T cell proportions. We also evaluated data from the on treatment timepoint. Here, we observed that prominent components could effectively separate integrated patient data by age (PC4, Fig. 6C) as well as ICI response (PC2, Fig. 6D). Here, PC4 is driven by a memory/effector T cell signature and is inversely correlated with a naïve T cell signature, and PC2 is representative of a myeloid cell signature. Overall, we find that naïve T cell proportions account for most age-related variance in our data set, and that cellular proportions have an outsized effect compared to cytokine levels, thought variances are present across multiple principal components likely due to the underlying heterogeneity of the clinical cohort.

## Discussion

Over the past decade, ICIs targeting PD-1/PD-L1, CTLA-4, and other molecules that regulate the immune response against cancer have transformed the treatment of a wide variety of cancer types, enhancing survival, and in some cases even leading to complete tumor regressions. Despite the high prevalence of cancer in aged patients (≥65 years), the aged population is underrepresented in prospective clinical

trials of ICIs, potentially because of a high rate of comorbidities in this population that preclude participation in interventional clinical trials. ICIs are thought to induce antitumor responses through the re-invigoration of antitumor immunity, a process contingent upon the function of the host immune system. Aged patients show evidence of reduced immune 'fitness', as evidenced by an increased susceptibility to infection, cancer, as well as some autoimmune diseases. While age-dependent effects on overall immune function and abundance of immune cell subsets have been well characterized, the immune features of aged cancer patients, and the interplay between age-related cellular changes and ICI treatment response is not fully understood.

Consistent with some prior reports, we find that ICI clinical outcomes are comparable between aged (≥65 years) and younger (<65 years) patients in a diverse pan-tumor cohort. However, aged patients exhibit distinct immune response features and dynamics both at baseline and in response to ICI treatment. Specifically, aged patients have lower proportions of naïve cytotoxic (TcN) and helper T (ThN) cells, B cells, and double negative T (DNT) cells, and a higher proportion of natural killer (NK) cells before and after ICI treatment. Aged patients also display diminished circulating cytokine responses after treatment. In addition to a lower overall pool of naïve T cells, naïve T cells from aged patients exhibit unique phenotype with higher

expression of immune checkpoint molecules and markers including PD-1 and TIGIT. These cells are antigen inexperienced, and their expression of these activation/exhaustion markers are far lower than the antigen experienced cell populations in our dataset, but this highlights a key difference between the pool of naïve T cells found in aged compared to younger patients. While these features may be consistent with some degree of baseline immune impairment as a result of chronic inflammation that accompanies aging, aged patients who respond to ICIs paradoxically displayed larger expansions of cytotoxic T cell populations with treatment compared to aged non-responders, while there was not a significant expansion in cytotoxic T cells in young ICI responders compared to young non-responders. Nonetheless, this unique phenotype in naïve T cells from aged patients may represent a T cell state primed to produce hypoactive T cells upon antigen recognition. While this cannot be tested using data from our observational study, future work examining these naïve T cell pools using experimental models is warranted.

Limitations of our study include the heterogeneity of our cohort, which included patients with diverse demographics and solid tumor types treated with an assortment of different PD(L)1-based treatments, as well as the relatively modest size of our sample cohort, although we note that the sample size is higher than previously reported for this type of aging-focused prospective clinical study. The lack of a significant association between patient age and PFS and OS may be attributed to the short follow-up period of the study coupled with the variety of tumor types represented in the cohort, each with differing expected clinical courses. In addition, we limited our detailed analyses correlating immune responses with therapy responses in patients using BOR/RECIST as PFS/OS would likely be unsuitable given the high degree of heterogeneity (multiple tumor types with variable disease courses) in our cohort. Further, analysis of the local tumor microenvironment would add additional information regarding the immune response to ICIs for each patient, which was not possible with our current study design. The heterogeneity of our study cohort may have limited our ability to detect less pronounced age-related differences by biasing analyses towards the null hypothesis as compared to analyses conducted on prospective clinical trials, which generally accrue a more homogeneous population. However, the "real-world" nature of our prospective clinical cohort is also a strength of our study, as the comparably high accrual of both aged and non-white patients in our cohort reflects the diversity of cancer care in clinical practice.

While other studies have explored the impact of circulating immune cells with features of senescence or the impact of memory-like cells in single cancer type patient populations[66–70], our results are among the first to prospectively and deeply phenotype the circulating immune cell populations and cytokine responses to ICI treatment, and to tie unique phenotypes to patient age across tumor types. Whereas many prior studies have focused on features of senescence in circulating immune cells, our analyses have focused in on major differences in the phenotypes of naïve T cell populations in young and aged patients. Further, our prospective study was able to provide an opportunity to analyze relationships between not only circulating immune cell populations, but also circulating cytokine levels and patient outcomes. We illustrate that the presence of these age-related phenotypes is not directly tied to ICI efficacy, but rather may underlie unique mechanisms at play in the immune response after therapy. These findings are consistent with a model in which ICIs targeting the PD-1 checkpoint may reverse T cell exhaustion against cancer antigens in aged individuals, providing insight into the maintenance of ICI efficacy in the aged population.

Although the overall clinical benefit of ICI treatment is maintained in aged patients, the divergent immune features observed in this study implies that tailored combination approaches may be needed to improve ICI outcomes for aged versus younger cancer patients. For example, the markedly reduced naïve T cell pool in aged patients both at baseline and after ICI treatment suggests that ICI combination strategies that are dependent upon the availability and function of naïve T cells may have reduced efficacy in aged patients. Therapeutic cancer vaccines are one emerging therapeutic strategy for enhancing responses to ICI that are thought to be dependent upon naïve T cells to induce or expand anti-tumor T cell clones, and yet current clinical efforts to enhance ICI responses with the addition of a therapeutic cancer vaccine are generally being conducted in patient populations without age cutoffs. If clinical trials of therapeutic cancer vaccines plus ICIs show overall clinical benefit, it will be essential to investigate whether the benefit of these agents extends into aged patients in subgroup analyses. Conversely, our finding that aged patients have enhanced expression of certain T cell exhaustion markers that are targets of emerging cancer immunotherapies, such as TIGIT, indicate that additional ICIs in development may be hypothesized to show enhanced benefit in aged individuals. Collectively, these insights into age-stratified mechanisms of ICI effects imply the utility of developing age-tailored immunotherapeutic approaches.

## Methods

### Study approval
The study protocol was approved by the Johns Hopkins Institutional Review Board (IRB #00267960), and all participants provided written informed consent before the blood samples and clinical data were collected.

### Study design
We are conducting an ongoing prospective observational study of patients with solid tumors who received ICI treatment as standard of care at Johns Hopkins University from May 2021 to the present. Patients included in this data analysis were enrolled from May 2021 to October 2022, and data censorship was set at 6 months from last consented patient included in the analysis (April 2023)[71]. Eligible patients were aged >18 years with pathologically confirmed solid tumors treated with ICIs consisting of anti-PD-1/PD-L1 blockade (nivolumab, pembrolizumab, atezolizumab, cemiplimab, durvalumab, and avelumab), combination ICI blockade with anti-PD-1 (nivolumab) and anti-CTLA-4 (ipilimumab), or in combination with targeted therapy or chemotherapy. All patients enrolled in the study had peripheral blood samples collected at baseline prior to initiation of the ICI. Subsequently, peripheral blood samples were collected at month 1, 2, 4, 6, and 12 as long as the patient was continued on ICI and if available. Information regarding tumor molecular biomarkers of ICI response including tumor mutational burden (TMB), PD-L1, and high microsatellite instability (MSI-H) and mismatch repair deficient (dMMR) status, was available for a subset of patients who had undergone testing with a commercially available molecular profiling assay (including but not limited to Tempus, Caris, and Foundation Medicine) as standard of care. PD-L1 classification (none, low, high) was based on the assay utilized. If a patient had multiple types of PD-L1 assessments such as tumor proportion score (TPS) and combined positive score (CPS), then the PD-L1 value used for classification was the type of PD-L1 assessment utilized for that specific tumor type per national guidelines. Sex of the patients was determined based on self-reporting by patients in the electronic medical record for those patients enrolled in the clinical trial. Discrete sex-based analyses were not performed as the size of the resulting sex-stratified cohorts were relatively small.

### Clinical definitions
Patient characteristics were abstracted from the electronic medical record (EMR) including age, sex, race/ethnicity, cancer histology, baseline autoimmune history, prior oncologic treatment history, treatment setting (advanced/metastatic vs neoadjuvant/adjuvant), and type and dates of ICI treatments. Cancer groups for GI, GU, UAD, skin, and other are defined in the Supplementary Information. IrAEs were

defined based on Common Terminology Criteria for Adverse Events version 5 (CTCAE v.5.0). The dates of onset, grade, and duration of irAEs were determined from review by a clinical researcher and confirmed by a medical oncologist reviewer. To select for the most clinically relevant irAEs, the irAE for each patient that was used for a data analysis was the highest grade irAE. Responses were based on RECIST v1.1 criteria and treating oncologists' documentation. Progression free survival (PFS) was defined as time from initiation of ICI on study to progression or death, and overall survival (OS) was defined as time from initiation of ICI on study to death. Time to irAE onset was defined as the time from initiation of ICI on study to the onset of the highest grade irAE.

## Plasma sample collection

Blood samples were obtained in heparinized syringes by standard phlebotomy technique and processed within 2 h of collection. For the isolation of plasma, blood was transferred into a 50 ml conical tube and placed in a centrifuge at 1800 x g. for 20 min with the brakes off. The plasma layer was removed and stored in 1 ml aliquots at −80 °C. Peripheral blood mononuclear cells (PBMCs) from the remaining blood were isolated using a standard LeucoSep tube technique. Briefly, blood diluted in equal parts of PBS was added to LeucoSep tubes preloaded with Ficoll-Paque. The tubes were then centrifuged for 20 min at 1800 x g with no brake. After the PBMC suspension was collected, the cells were washed in PBS and stored in a cryovial initially at −80 °C before transfer to liquid nitrogen for long-term storage.

## Cytokine measurements

The Bioplex 200 platform (Biorad, Hercules CA) was used to determine the concentration (pg/mL) of multiple target cytokines in plasma. Luminex bead-based immunoassays (Millipore, Billerica NY) were performed following the Johns Hopkins Immune Monitoring Core SOPs and concentrations were determined using 5 parameter log curve fits (using Bioplex Manager 6.0) with vendor-provided standards and quality controls. The HCYTA-60K panel was used to detect 32 cytokines (sCD40L, IL-1α, IL-1β, IL-1Ra, IL-2, IL-3, IL-4, IL-5, IL-6, IL-8, IL-9, IL-10, IL-12p40, IL-12p70, IL-13, IL-15, IL-17a, IL-17f, IL-18, IL-22, IL-25, G-CSF, GM-CSF, TNF-α, IFN-γ, CCL2 (MCP-1), CCL3 (MIP-1α), CCL4 (MIP-1β), CCL5 (RANTES), CXCL9 (MIG), CXCL10 (IP-10), VEGF-α). Concentrations which were outside of the standard curves values were categorized as "out of range" (OOR). For each cytokine, OOR < values lower than the lower limit of detection were replaced with the lower limit of the standard curve of the assay while OOR > values greater than the upper limit of detection were replaced with the upper limit of the standard curve. To account for batch effect and ensure accurate fold change calculations, on-treatment samples of each patient were always run with the corresponding baseline sample. For patients with baseline samples that were run in multiple batches, the average concentration of those baseline samples was used in baseline analyses.

## Antibodies

All antibodies used for CyTOF are listed in a table included in Supplementary Data 1. Custom antibodies were conjugated as previously described in ref.[72].

## CyTOF staining, acquisition, and analyses

Patient PBMCs were thawed in a 37 °C water bath and gradually recovered using pre-warmed RPMI 1640 media containing 10% fetal bovine serum (FBS). Samples were then counted and $2 \times 10^6$ cells from each sample were plated in 96-well plates. Cells were allowed to rest in the media for 30 min prior to staining. Cells were then washed once in PBS with 2 mM EDTA. Next, cells were incubated for 2.5 min at room temperature in 20 uM Pt (Standard BioTools) in PBS to mark for viability. Following the 2.5 min, RPMI containing 10% FBS was added to

the cells to quench any residual platinum. This is followed by two washes with cell staining buffer (CSB) (Standard BioTools). Following washes, all samples were barcoded by incubating the cells with unique combinations of metal conjugated anti-CD45 antibodies for 20 minutes. After 2 washes with CSB, samples were multiplexed and transferred to v-bottom flow tubes using a 40 um filter. Each tube is then blocked using anti-human FcR block (12 μl used for $15 \times 10^6$ cells) for 10 min at room temperature. This was followed by a chemokine stain cocktail for 10 min in a 37 °C water bath. Tubes were removed from the water bath and a surface stain cocktail (Supplementary Data 1) was added for 30 min at room temperature. Samples were washed twice with CSB and then fixed and permeabilized using cytofix/cytoperm solution (BD Biosciences) for 30 min at room temperature. Fixed and permeabilized samples were then washed with perm/wash solution (BD Biosciences) and then subsequently stained using the intracellular cocktail for 30 min at room temperature. Samples were washed twice with perm/wash solution and stored in 1.6% PFA in PBS at 4 °C until the day of acquisition, not exceeding one week. On the day of acquisition, samples were stained with 1:500 $^{103}$Rh in Maxpar Fix/Perm solution (Standard BioTools) for 30 min at room temperature for cell identification. Samples were washed with PBS once and then washed and resuspended in normalization beads (Standard BioTools). The data were acquired on a Helios mass cytometer (Standard BioTools) at the Johns Hopkins University Mass Cytometry Facility. All acquired data was randomized and normalized using CyTOF software (v7.1.16389.0, Standard BioTools). Resulting fcs files were then debarcoded by manual gating using FlowJo software (v10.9.0, BD Biosciences). Cell events were gated using $^{103}$Rh positivity. Live cells were then gated based on $^{194}$Pt and $^{195}$Pt viability staining. This was followed by debarcoding based on positivity of unique combinations of CD45 barcodes. Each debarcoded sample was then exported as an individual fcs file. Samples were normalized in R (v4.0.2) using the CytoNorm algorithm that utilized a repeated sample included in each staining batch to normalize all samples based on goal quantiles of mean marker expression[73]. Clustering analysis was performed in R using FlowSOM to generate 35 metaclusters that were annotated using the expression profile of markers included within the panel, resulting in 26 final clusters (Supplementary Data 2)[74]. The PD-1 antibody used in the CyTOF analysis competes with anti-PD-1 therapies received by patients in this study. Therefore, we did not consider PD-1 expression for any on treatment timepoint analyses.

## Statistical methods

For samples collected on treatment, fold change for each timepoint was calculated relative to baseline to account for inter-patient variability. For visualization purposes, $\log_2$ transformation to measurements or fold changes of cytokines or CyTOF data were utilized. For constructed heatmaps when specified, scaling was employed to normalize inter-group heterogeneity for different cytokines and immune clusters. Scaling was performed by calculating the z-score of each individual patient for each cytokine or immune cluster of interest. Z-scores represent the number of standard deviations from the cohort mean and is calculated by the following formula (patient measurement − cohort mean of measurement)/(cohort standard deviation of measurement). Survival outcome differences between groups were compared using Kaplan-Meier curves and differences in cumulative probability of irAEs between groups was assessed using reverse Kaplan-Meier curves; both calculated P values using the log-rank test.

The Wilcoxon rank-sum test was used when assessing statistical differences between two groups, and the Kruskal-Wallis test was used for three groups or higher. A Fisher's exact test was utilized when assessing statistical differences between two categorical variables. Box and whisker plots were utilized to show visually the median,

interquartile range, minimum, and maximum. Multivariable analyses via linear regression models were conducted to assess the associations between cytokines or immune clusters and age, adjusting for cancer type (GU vs. GI, and Others vs. GI) and prior oncologic systemic therapy (Yes vs. no). These adjustments were selected based on the distribution of patient across age cohorts and potential confounders identified with possible marginally significant associations. Without loss of generality given that linearity assumptions were met for age in all cytokines and most of immune clusters (except TcN at baseline and on-treatment time points but displayed monotonic trends by age), sensitivity analyses treating age as a linearly continuous variable were conducted. Specifically, the linearity of age was tested using regression models that incorporated restricted cubic spline function with three knots placed at the 25th, 50th, and 75th percentiles of age. Furthermore, the effects of age cohort on cytokines or immune clusters in relation to treatment response were evaluated by including interaction terms in the regression models, and the significance of these interaction effects were assessed accounting for multiplicity adjustments[75]. All statistical tests were two-sided and considered significant at $P$-value $< 0.05$ unless stated otherwise. PCA was performed in R. To facilitate cross-comparison of disparate datasets, the data was first normalized to a range of 0 to 1 based on the 0th to 95th percentile of each measured parameter. The principal components that best separated the data by age or response were selected for representative biaxial plotting by comparing the differences in average PC scores. Statistical analyses were performed using RStudio software (Version: 2023.03.0 + 386).

### Reporting summary

Further information on research design is available in the Nature Portfolio Reporting Summary linked to this article.

## Data availability

De-identified CyTOF data files will be deposited to and available through Zenodo[76]. The authors declare that the minimal data set for this study cannot be shared publicly due to ethical and legal restrictions on sharing de-identified data that aligns with the consent of research participants. Current JHU compliance policies require data with no direct consent for public open access sharing be under restricted access. We will provide access through Vivli, an established repository for clinical data that provides open access without a fee restricted to approved researchers under a Data Use Agreement. JHU compliance policy for Vivli requires additional anonymization of certain demographics, including use of age ranges and limiters to outlier values for weight, height, and certain rare diseases, while retaining sufficient value for reference and validation of results. Researchers can request more detailed data from the corresponding author shared though an approved collaboration arrangement. Access to some data may be restricted due to patient privacy protection concerns. Source data are provided with this paper. The remaining data are available within the Article, Supplementary Information, or Source Data file. Source data are provided with this paper.

## Code availability

Custom code used to generate the results in this study has been deposited in a GitHub repository at https://github.com/dzabran1/ICI-Aging[77] and https://github.com/wjhlab/Age_Related_CyTOF[78].

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

## Acknowledgements

We thank the patients and their families and all the investigators who contributed to the collection of biospecimens used in this study. Funding acknowledgements include R01CA197296-06 (E.M.J), R01CA293602-01 (E.M.J.), P01CA247886-01A1 (E.M.J.), Johns Hopkins Bloomberg-Kimmel Institute for Cancer Immunotherapy (M.Y.), the NCI SPORE in Gastrointestinal Cancers (P50 CA062924) (M.Y.), the NIH Center Core Grant (P30 CA006973) (M.Y.), Swim Across America (M.Y.), imCORE-Genentech grant 137515 (to Johns Hopkins Medicine on behalf of M.Y.), S10OD034407 (W.J.H.), NIH/NCI R21CA264004 (W.J.H.). the MacMillan Pathway to Independence Award (D.J.Z.), the MD Anderson GI SPORE Career Enhancement Award NCI P50 CA221707 (D.J.Z.), and the Maryland Cancer Moonshot Research Grant to the Johns Hopkins Medical Institutions (FY24) (D.J.Z.).

## Author contributions

Conceptualization: C.K., M.Y., D.J.Z. Methodology: C.K., H.T., W.J.H., M.Y., D.J.Z. Software: C.K., S.C., H.T., W.J.H. Formal analysis: C.K., S.C., H.T., W.J.H., M.Y., D.J.Z. Funding Acquisition: E.M.J., W.J.H., M.Y., D.J.Z. Investigation: C.K., S.C., H.T., K.A., D.H.S., K.M., E.G., J.M.L., H.L.L., E.H., M.B., M.N., S.A., C.T., N.E.G., A.G.H., E.M.C., E.K., M.B., S.B., L.T., G.S.C., R.M. Resources: E.J.L., Y.G., J.H-C., M.B., W.J.H., M.Y. Supervision: W.J.H., M.Y., D.J.Z. (lead). Writing – Original Draft Preparation: C.K., D.J.Z. Writing – Review & Editing: All authors.

## Competing interests

E.J.L. reports the following: Consultant/Advisor: Agenus, Bristol-Myers Squibb, CareDx, Eisai, Genentech, HUYA Bioscience International, Immunocore, Instil Bio, IO Biotech, Lyvgen, Merck, Merck KGaA, Natera, Nektar, Novartis, OncoSec, Pfizer, Rain Therapeutics, Regeneron, Replimune, Sanofi-Aventis, Sun Pharma, Syneos Health. Institutional Research Funding: 1104Health, Bristol-Myers Squibb, Haystack Oncology, Merck, Regeneron, Sanofi. Stock: Iovance. E.J.L is supported by Bloomberg~Kimmel Institute for Cancer Immunotherapy, the Marilyn and Michael Glosserman Fund for Basal Cell Carcinoma and Melanoma Research, the Barney Family Foundation, and the Laverna Hahn Charitable Trust. M.B. reports Grant/Research Support (paid to Johns Hopkins): Merck and Consulting: Exelixis, Incyte, AstraZeneca. E.M.J. reports other support from Abmeta and Adventris, personal fees from Dragonfly, Neuvogen, Surge Tx, Mestag, HDTbio, and grants from Lustgarten, Genentech, BMS, NeoTx, and Break Through Cancer. Dr. Elizabeth Jaffee is a founder of and holds equity in Adventris Pharmaceuticals. She also serves as a consultant to the entity. Further, Adventris Pharmaceuticals has licensed a technology described in this study from the Johns Hopkins University. As a result of that agreement, Dr. Jaffee and the University are entitled to royalty distributions related to technology described in the study discussed in this publication. This arrangement has been reviewed and approved by the Johns Hopkins University in accordance with its conflict of interest policies. G.S.C. is an employee and stockholder of Roche. G.S.C. received support for preparation of this manuscript and corresponding travel from Roche. G.S.C. is a co-inventor on patents filed by Genentech/Roche that are related to atezolizumab use. R.M. is an employee and stockholder of F. Hoffman-La Roche, Ltd. R.M. is a co-inventor on patents filed by Genentech/Roche that are related to atezolizumab use. W.J.H. reports patent royalties from Rodeo/Amgen; grants from Sanofi, NeoTX, and Riboscience; speaking/travel honoraria from Exelixis and Standard BioTools. M.Y. has received grant/research support (to Johns Hopkins) from Bristol-Myers Squibb, Incyte and Genentech/Roche; has received honoraria from Genentech/Roche, Exelixis, AstraZeneca, Replimune, Hepion, and Lantheus; and is a cofounder with equity of Adventris. D.J.Z. reports grant support (to Johns Hopkins) and travel from Roche/Genentech, and honoraria from Omni Health Media, Sermo, Atheneum, Escientiq, Deerfield Institute, ZoomRx, and NeuCore Bio. The remaining authors declare no competing interests.
