## [Transparent Peer Review file · Nature Communications]

Age-related divergence of circulating immune responses in patients with solid tumors treated with immune checkpoint inhibitors

Corresponding Author: Dr Daniel Zabransky

Version 0:

Reviewer comments:

Reviewer #1

(Remarks to the Author)

Age-related divergence of circulating immune responses in patients with solid tumors treated with 2 immunotherapy

In this article, Kao et, al. characterised immune landscape of patients treated with ICI, at baseline and on-treatment, and intended to seek age-related divergence that are associated with ICI treatment response. There has been many efforts trying to characterize the age-related landscapes on cellular, genetic, epigenetic and proteomic levels, as well as the drifting of such landscapes in tumorigenesis and response to treatment. This study will certainly improve our understanding on these topics.

My biggest concern is the quality of data analysis, which is not sufficient yet to underpin the conclusion of this article. Clearly, the samples used in this study are from a very heterogenous patient cohort. While the authors tried to build a three-way linkage among age, ICI response, and immune landscapes (cytokines, cell types, surface, etc), many clinical factors including tumor types, stages, treatment history may act as confounders. So far, the authors have primarily used univariable analysis such as K-M curve and K-W statistical tests to explore the pair-wise relationship between age, ICI response, and immune landscapes, without a systematic solution to exclude the confounding effect introduced by the heterogenous patient cohort. Moreover, the dichotomisation of patients by the age cut-off of 65 is questionable and may weakened the significance that this study could achieve.

1. In Table 1, there is an attempt to demonstrate that age is independent of all confounding clinical factors. However, some association are marginally significant, and the authors needs to seek external evidence to further support, or challenge, the conclusion of independence they made based on this dataset alone.
2. Additional multivariable analysis adjusting for the confounding clinical variables will be needed in order to explore the three-way relationship among age, ICI response, and immune landscapes.
3. It is a surprise that in Fig1b the KM does not reached significance. I would blame the lack of sufficient sample size and would highlight the need of additional multivariable analysis to understand the observed difference in PFS.
4. I noticed that the difference between young and old cohort are moderate even if statistical significance (without correction for multiple comparison) were reached in some cases (Figure 2, ext Fig 2, 3, etc). This is partly due to the use of 65 as a cutoff. In fact, it is hard to believe that 65 of age is a biological meaningful cutoff. I recommend treating age as a continuous variable before applying the cutoff, or use age<33 quantile vs. age>66 quantile to try to observe more contrasting age-related difference in biology
5. The authors identified unique cell phenotype related to age. Again, could this signal be driven by clinical factors such as tumour type?

(Remarks on code availability)

Reviewer #2

(Remarks to the Author)

The manuscript provides a comprehensive analysis of peripheral immune changes following treatment with immune checkpoint inhibitors (ICIs), focusing on peripheral cytokines and cellular composition. Overall, the concept of identifying peripheral immune features that predict response to ICI treatment is compelling, and the significant impact of age on immunity suggests a strong age interaction that is significantly considered in the manuscript. However, the current analytical pipeline seems insufficiently thorough and lacks an integrative approach, which could better utilize the data. Specifically, the authors analyze each data type (cytokines and cells) separately, missing the opportunity to consider the data as an integrated whole, thereby masking potential correlations and interactions between the features.

Here are the major points that should be revised to improve the manuscript, primarily in terms of the analytical pipeline:

1. Cohort Characterization: Can the authors assess whether there is an association between immune-related adverse events (irAE) and response to treatment within the cohort (as known from the literatures)? In general, I suggest expanding the outcome-associated analyses included in the manuscript to provide prognostic immune features relevant to both outcome measurements and not only response to treatment.
2. Cytokine Baseline Levels as a Function of Age and Tumor Type:
 - a. I suggest conducting continuous analysis of cytokine expression versus patient age to potentially reveal more significant results, instead of single-cytokine analysis between two age groups.
 - b. Numerous studies exist that include correlations between peripheral cytokines and solid tumors, and peripheral cytokines and age – can the authors compare their results against these studies' results?
 - c. I suggest analyzing groups of cytokines rather than individual cytokines to gain functional insights and enhance signals by reducing noise.
 - d. I suggest classifying cytokines based on their association with age or tumor class – the authors may use an integrative model that takes both tumor type and age as predictors to cytokines levels.
 - e. The age threshold defining older and young adults seems to be arbitrarily chosen. Can the authors perform a sensitivity analysis of the threshold defining young versus old patients? This is true also for cellular abundance analysis that appears below.
 - f. There is available data of peripheral blood scRNA sequencing in solid tumors – can the authors use it as a reference to identify which cells are responsible for the secreted cytokines in different solid tumors? These can be correlated with the cell subsets measured by CyTOF.
3. Cytokine Responses to Treatment:
 - a. Can the authors provide an overview of which cytokines respond to ICI treatment across the entire population and within specific cancer types, irrespective of age?
 - b. Based on the above, the authors can assess whether the above response patterns are associated with clinical outcomes, such as survival, and examine any dose-response relationship between ICI treatment intensity (i.e. single vs. dual anti ICI therapy) and typical cytokine response.
 - c. Cluster patients based on their cytokine responses and correlate these groups with age, outcome, and irAE.
 - d. Explore relationships between immune-therapy response parameters (e.g., TMB, MSI) and cytokine baseline and response.
4. Cellular Abundance:
 - a. In Figure 3C the authors may derive better correlations by regressing cell type abundance against the actual age of patients, rather than calculating differences between age groups. If available, CMV data may be included due to its significant impact on baseline immune variability, particularly in older individuals.
 - b. Also here, the authors may investigate the correlation between cellular abundance and tumor type, as observed for cytokines.
 - c. I strongly suggest calculating cellular abundances of subsets as proportions of parent populations instead of the entire cell population for more biologically meaningful analysis (e.g., naive CD8+ T cells as a fraction of CD8+ T cells).
 - d. As for the cytokines analysis mentioned above, I suggest analyzing and providing the typical cellular abundance changes following ICI treatment across age groups, identifying which changes are enriched in specific age populations or outcomes (e.g., responders vs. non-responders, irAE).
5. Dimensionality and Context: Since all the measurements provided by the manuscript (cytokine levels and cellular abundances) were measured in the same individuals, the authors may provide a low-dimensional representation of patients based on cell subsets abundance and cytokines, showing the positioning of aged versus young individuals, before and after ICI treatment. Can the authors identify a specific axis along which individuals are changing their immune composition over time, thus determining the direction most individuals move toward after treatment in this low-dimensional space. Is this axis associated with outcomes measurements such as irAE and outcome?
6. Can the authors propose an integrative model using baseline cytokines, age, cellular data, and fold-changes after ICI treatment to predict clinical features? This may be presented a correlation network between cells and cytokines measured in the same individuals, showing how edges change with age and treatment. Highlight how clusters of cells and cytokines predict treatment response.
7. From a statistical perspective, I suggest ensuring that the comparisons throughout the manuscript are corrected for multiple hypotheses.
8. Biological Age Stratification: Aged individuals are heterogeneous. I suggest checking the associations identified between cellular abundances and cytokines and outcome to be compared against biological age (determined by baseline immune feature levels) instead of chronological age.

(Remarks on code availability)

(Remarks to the Author)

Peer review comments for "Age-related divergence of circulating immune responses in patients with solid tumors treated with immunotherapy" by Chester Kao et al., for Nature Communications

Summary

The manuscript "Age-related divergence of circulating immune responses in patients with solid tumors treated with immunotherapy" profiles cancer patient peripheral cytokine levels and immune cell subsets. They analyze the data using the Bioplex 100 platform, Luminex bead-based immunoassays, and CyTOF, and compare differences between younger and aged individuals in a diverse cohort of many cancer types and assess how this impacts response to immune checkpoint inhibitor therapy. Through this method, the authors report that aged patients have differences in immune phenotypes at baseline and on treatment, including reduced levels of naïve T cell with increased immune checkpoint protein expression, as well as an enhanced effector T cell expansion in older responders compared to older non-responders. This study provides an important contribution to the field, however there are several key points that need to be addressed.

Major points

More details are needed on prior oncologic treatment, as the immune profile can be greatly impacted by prior therapies (type, # of lines, etc); some of what is described here may have nothing to do with age, but rather the prior treatment history of the patients evaluated in this study.

Associations between circulating immune subsets and clinical outcome only include associations with BOR. Did the authors also perform any analyses with PFS or OS as an endpoint? Additional key features may be uncovered that do not associate with BOR but do associate with longer term outcomes of patients.

Although there were no age associated differences at baseline in circulating analytes (cytokines), were there any associations between circulating analytes and clinical outcomes (either in the combined age cohort, the old cohort, the young cohort, or in patients with HCC or RCC (your two biggest cancer type cohorts)? In addition, since there were cancer type associated differences in cytokines shown in Extended Data Figure 1, were there any age-related differences in cytokines when broken down into HCC or RCC (your two biggest cancer type cohorts)?

When were post treatment bloods collected and analyzed? In the results, you state that blood was collected approximately 1-6 months after initiation of a checkpoint inhibitor therapy (line 105). In the methods, you state that peripheral blood samples were collected at month 1, 2, 4, 6, and 12 (line 393). In the figures, the timepoint is just referred to as "on-treatment". Which timepoint(s) was(were) used in the analysis? A 6th month window is likely too wide of a window post treatment and should be narrowed, even if the n for paired analyses pre and post treatment is much smaller. Many changes in circulating analytes, for example that are seen a month after initiating immune checkpoint blockade are no longer seen after multiple cycles of treatment, which may impact the results of the current study. If the window of 6 month is kept, you really need to add this as a variable in the heatmap of anything where you are looking at post or on-treatment timepoints.

Do any changes in circulating analytes after therapy associate with clinical outcomes (either in the combined age cohort, the old cohort, the young cohort, or in patients with HCC or RCC (your two biggest cancer type cohorts)?

It should be noted that many of the immune cell subset differences described between young and old patients in this study are not novel (e.g. naïve T cell subsets, NK cells).

Do the detection antibodies utilized to detect PD-1, PD-L1, CTLA4 with the CyTof assay compete with the checkpoint therapies some of these patients have received previously, or are treated with in the current study? This should be considered in interpreting data from any patients post treatment with checkpoint blockade.

Line 275. Did you perform any comparisons of responders vs non-responders for the analysis of immune checkpoints? Since aged patients have increased expression of immune checkpoints in their naïve T cells, how does this specifically relate to response to therapy, for example?

Was an age cutoff other than 65 considered for this work?

Many of the figures are not legible when printed – the text is far too small and blurry which makes the study extremely difficult to interpret/review. The following figures especially need to be enlarged/improved: Figure 2B (heatmap), Figure 3C and D (graphs and text too small), Figure 4 – text too small, all graphs and heatmaps in Figure 5 (can't read anything).

Minor points

Last line of the introduction – starting on line 94 is an incomplete sentence – needs to be re-worded.

More details on PD-L1 classification is needed.

Line 378. Suggest ending with conclusions instead of limitations.

Running title should be changed to Cellular responses; there are no Molecular responses being evaluated in the current study.

Supplemental tables need to be combined so all columns are on a single page – can't make sense of them as they are now – spread out on multiple pages

When discussing post treatment analytes, please be clear whether absolute levels post treatment or changes compared to baseline are being evaluated.

(Remarks on code availability)
No code provided.

Reviewer #4

(Remarks to the Author)
I co-reviewed this manuscript with one of the reviewers who provided the listed reports. This is part of the Nature Communications initiative to facilitate training in peer review and to provide appropriate recognition for Early Career Researchers who co-review manuscripts.

(Remarks on code availability)

Version 1:

Reviewer comments:

Reviewer #1

(Remarks to the Author)
The manuscript has been substantially revised. With the additional multivariable analysis and including age as both a continuous and a categorical variable, the three-way relationship among age, cytokines (or clinical variables, immune cell composition, checkpoint markers, etc) and treatment outcomes have been clarified. I am largely convinced now on the main conclusions, and was pleased that the drawbacks on data size, heterogeneity, etc have been properly discussed in the manuscript.

(Remarks on code availability)

Reviewer #2

(Remarks to the Author)
I co-reviewed this manuscript with one of the reviewers who provided the listed reports. This is part of the Nature Communications initiative to facilitate training in peer review and to provide appropriate recognition for Early Career Researchers who co-review manuscripts.

(Remarks on code availability)
The authors have provided satisfactory responses to my concerns and have adequately addressed all raised issues. I appreciate their effort and thoughtful revisions. Based on the improvements made, I recommend the paper for publication.

Reviewer #3

(Remarks to the Author)
This manuscript has been substantially improved since the prior version.

Please see attached file detailing a few additional minor points that still need to be addressed by the authors.

(Remarks on code availability)
I have not assessed the code.

Reviewer #4

(Remarks to the Author)
I co-reviewed this manuscript with one of the reviewers who provided the listed reports. This is part of the Nature Communications initiative to facilitate training in peer review and to provide appropriate recognition for Early Career Researchers who co-review manuscripts.

(Remarks on code availability)

RESPONSE TO REVIEWER COMMENTS

Reviewer #1 (Remarks to the Author): with expertise in biostatistics, biomarkers, cancer

Age-related divergence of circulating immune responses in patients with solid tumors treated with 2 immunotherapy

In this article, Kao et, al. characterised immune landscape of patients treated with ICI, at baseline and on-treatment, and intended to seek age-related divergence that are associated with ICI treatment response. There has been many efforts trying to characterize the age-related landscapes on cellular, genetic, epigenetic and proteomic levels, as well as the drifting of such landscapes in tumorigenesis and response to treatment. This study will certainly improve our understanding on these topics.

My biggest concern is the quality of data analysis, which is not sufficient yet to underpin the conclusion of this article. Clearly, the samples used in this study are from a very heterogenous patient cohort. While the authors tried to build a three-way linkage among age, ICI response, and immune landscapes (cytokines, cell types, surface, etc), many clinical factors including tumor types, stages, treatment history may act as confounders. So far, the authors have primarily used univariable analysis such as K-M curve and K-W statistical tests to explore the pair-wise relationship between age, ICI response, and immune landscapes, without a systematic solution to exclude the confounding effect introduced by the heterogenous patient cohort. Moreover, the dichotomisation of patients by the age cut-off of 65 is questionable and may weakened the significance that this study could achieve.

1. In Table 1, there is an attempt to demonstrate that age is independent of all confounding clinical factors. However, some association are marginally significant, and the authors needs to seek external evidence to further support, or challenge, the conclusion of independence they made based on this dataset alone.

We thank the reviewer for their assessment and comments regarding the manuscript. In Table 1, we present the clinical characteristics of the patients enrolled in our study. We do not argue that age is independent of confounding clinical factors in this table, but with a comparison of our young (<65) and age (≥65 year old) patient groups we did not find any statistically significant differences in these clinical factors between the groups.

However, to the reviewer's point, understanding if age independently impacts the studies we have performed, namely analysis of circulating cytokines and CyTOF for identification of circulating immune cell populations/abundances, is important. Therefore, we incorporated multivariable models, adjusting for cancer group (GU vs. GI, and Others vs. GI) and prior oncologic systemic therapy (Yes vs. No). These adjustments were selected based on the distribution of patient numbers across groups in Table 1, as well as the factors with possible marginally significant associations (smaller p-values in Table 1) indicating potential confounding factors. irAE status is the closest to having a statistically significant p-value here, but we did not adjust for this as we believe it to be more of a clinical outcome than patient demographic/characteristic.

These multivariable analyses are presented in **Supplementary Figure 7** and **Supplementary Figure 14**. Overall, we find similar trends to our initial univariate analysis, suggesting that the baseline demographic factors are not the major drivers of the findings of our study.

2. Additional multivariable analysis adjusting for the confounding clinical variables will be needed in order to explore the three-way relationship among age, ICI response, and immune landscapes.

We thank the reviewer for this suggestion. We have included newly performed multivariable analyses (assessing the impact of age on cytokine/and immune cells abundance dynamics in relation to treatment response which is now included as **Supplementary Figure 9** (cytokine) and **Supplementary Figure 16** (CyTOF).

Overall, we find strikingly similar results to the univariate analysis, which we believe highlights that much of the overall signal we've observed is driven by patient age. In the manuscript, we highlight that multivariable and interaction testing has again identified key difference in baseline levels of naïve T cell populations (lower in aged patients), and in CCL2 expression. A notable finding that is now highlighted in the revised manuscript is that

while aged patients have diminished fold changes in CCL2 expression compared to young patients, aged responders had a higher fold change in CCL2 compared to aged non-responders, and this trend was the opposite in young patients (response is associated with lower fold change in CCL2 in young patients). As CCL2 targeted immunotherapies are being studied as possible therapies to use in combination with ICIs, highlighting these divergent responses/associations may be helpful in future studies evaluating the patients most likely to respond to new immunotherapies.

3. It is a surprise that in Fig1b the KM does not reached significance. I would blame the lack of sufficient sample size and would highlight the need of additional multivariable analysis to understand the observed difference in PFS.

We acknowledge that the lack of significance may be attributed to insufficient power. In our revised manuscript, this limitation is stated in the manuscript. In addition, we agree that a multivariable approach could provide more understanding of relationship and we also included the related results in our revised manuscript as described above.

4. I noticed that the difference between young and old cohort are moderate even if statistical significance (without correction for multiple comparison) were reached in some cases (Figure 2, ext Fig 2, 3, etc). This is partly due to the use of 65 as a cutoff. In fact, it is hard to believe that 65 of age is a biological meaningful cutoff. I recommend treating age as a continuous variable before applying the cutoff, or use age<33 quantile vs. age>66 quantile to try to observe more contrasting age-related difference in biology

We thank the reviewer for noticing these trends and suggesting this analysis. Other reviewers also had questioned our use of 65 as an age cutoff to define the patient groups. We have conducted multivariable analyses with age as a continuous variable in cytokines and CyTOF outcomes. Consequently, the results of treating age as a continuous variable were overall consistent as our primary findings in this manuscript focus on using age 65 as the cutoff. We have included these additional analyses in the revised manuscript, which offers a broader perspective on age-related differences and helps readers to mitigate concerns regarding the potentially arbitrary nature of the initial age cutoff.

For further context, we opted to classify patients as “younger” or “aged” based on the 65-year threshold for several reasons: (1) Age 65 is a commonly used threshold to define older individuals in the U.S., and it aligns with SEER cancer statistics illustrating that the median age of cancer diagnosis in the U.S. is 65-67 (depending on the year) (seer.cancer.gov); (2) The median age of our cohort was 65; (3) This cutoff facilitates clearer interpretation of the results; (4) We observed a reasonable assumption of linearity in the age-cytokine relationship; (5) There is growing evidence from recent studies focused on biological changes that occur with aging that identify aged 65 as a unique chronological age at which multiple age-driven changes in cellular phenotypes (including RNA expression in circulating immune cells) become evident (doi:10.1038/s43587-024-00692-2 and doi:10.1101/2024.09.10.612119).

5. The authors identified unique cell phenotype related to age. Again, could this signal be driven by clinical factors such as tumour type?

One of findings was a decreased baseline pool of circulating naïve T cell populations in aged patients that remained lower than in young patients post-ICI treatment (**Figure 3C-D**). We performed multivariable analysis which incorporated additional clinical factors, notably tumor type and prior treatment with oncologic therapies and found similar results (**Supplementary Figure 14**). Even when adjusting for these additional clinical factors, aged patients had a distinct phenotype in terms of their circulating immune cell populations. With respect to exhaustion marker expression on naïve T cell clusters, we have included additional explorations of the mean metal intensity for functional markers in TcN cells in **Supplementary Figure 18**. We see a consistent trend across aged patients for higher PD-1 expression in TcN cells.

Reviewer #2 (Remarks to the Author): with expertise in system immunology

The manuscript provides a comprehensive analysis of peripheral immune changes following treatment with immune checkpoint inhibitors (ICIs), focusing on peripheral cytokines and cellular composition. Overall, the

concept of identifying peripheral immune features that predict response to ICI treatment is compelling, and the significant impact of age on immunity suggests a strong age interaction that is significantly considered in the manuscript. However, the current analytical pipeline seems insufficiently thorough and lacks an integrative approach, which could better utilize the data. Specifically, the authors analyze each data type (cytokines and cells) separately, missing the opportunity to consider the data as an integrated whole, thereby masking potential correlations and interactions between the features.

Here are the major points that should be revised to improve the manuscript, primarily in terms of the analytical pipeline:

1. Cohort Characterization: Can the authors assess whether there is an association between immune-related adverse events (irAE) and response to treatment within the cohort (as known from the literatures)? In general, I suggest expanding the outcome-associated analyses included in the manuscript to provide prognostic immune features relevant to both outcome measurements and not only response to treatment.

Thank you for this insightful suggestion. We examined the association between immune-related adverse events (irAEs) and treatment response within our young and aged patient groups. In summary, patients who responded to treatment were more likely to develop irAEs than those who did not respond, although these results did not reach statistical significance. We now present this in the revised manuscript as **(Supplementary Figure 1)**. Further, aged responders were more likely to develop irAEs than young responders, though again this was not statistically significant. These comparisons are limited by overall sample size, and thus we have chosen to concentrate on the more robust findings presented in our current analyses, which we believe best align with the manuscript's objectives. A goal of our study is to highlight the associations that age-driven – clinical, cytokine levels, or cellular abundances in the baseline and ICI exposed settings. As our initial exploration revealed no strong associations between clinical outcomes (such as PFS, OS) and patient age in this cohort, we did not expand outcome association analysis further to evaluate the prognostic immune features related to age.

*2. Cytokine Baseline Levels as a Function of Age and Tumor Type:
a. I suggest conducting continuous analysis of cytokine expression versus patient age to potentially reveal more significant results, instead of single-cytokine analysis between two age groups.*

We have extended our analysis for assessing the correlations between cytokine expression and age treating as a continuous variable. Please refer to reviewer 1, item 4, and with details listed in the methods section.

b. Numerous studies exist that include correlations between peripheral cytokines and solid tumors, and peripheral cytokines and age – can the authors compare their results against these studies' results?

We have added a brief discussion of this point into the revised manuscript. Particularly interesting is that we did not observe baseline differences in cytokine expression levels based on patient age. Prior studies have illustrated changes in baseline cytokine levels with age, however, most aging focused studies make comparisons across a larger spectrum of the lifespan compared to what we do in our study, as most patients in our cohort are over the age of 45 (such as <https://doi.org/10.1016/j.cellimm.2012.01.001>). Our study is unique in that we have influences of both cancer and aging, and we are able to describe unique features of relatively older vs. younger patients with cancer in terms of their cytokine responses to ICI treatment.

c. I suggest analyzing groups of cytokines rather than individual cytokines to gain functional insights and enhance signals by reducing noise.

We thank the reviewer for the suggestion to perform this interesting analysis. We performed a summed Z-score analysis for classes of cytokines that are denoted in Figure 2B (for example Th1, Th2, Th17, etc.). We did not observe any significant differences between the age group except in lower fold change differences in Treg in aged patients, though this was driven solely by IL-10 (**Supplementary Figure 4**). As most of the measured cytokines have multiple functions and can be context dependent in their action, our groupings may be too rudimentary to fully capture any true signal. Therefore, we favor comparisons of individual cytokine levels between young and aged patients at this time.

d. I suggest classifying cytokines based on their association with age or tumor class – the authors may use an integrative model that takes both tumor type and age as predictors to cytokines levels.

We found this to be a difficult task as most of our measured cytokines have been associated with multiple tumor types that are present in our cohort, and thus prespecifying which cytokines have previously been associated with a particular tumor type would result in most cytokines being associated with all of our cancer types in this study. We therefore performed multivariable analysis (as mentioned above) that takes tumor type and age into account. The results were overall similar to our initially reported univariate results.

e. The age threshold defining older and young adults seems to be arbitrarily chosen. Can the authors perform a sensitivity analysis of the threshold defining young versus old patients? This is true also for cellular abundance analysis that appears below.

Thank you for this suggestion. We discuss this issue in our response to Reviewer 1, item 4 above.

f. There is available data of peripheral blood scRNA sequencing in solid tumors – can the authors use it as a reference to identify which cells are responsible for the secreted cytokines in different solid tumors? These can be correlated with the cell subsets measured by CyTOF.

In the timeframe allotted for this revision, we focused on analysis of our cohort, but fully agree that understanding the source of the cytokine differences is of great interest. To that end, we performed correlation matrix analysis of cellular abundances of peripheral immune cells as measured by CyTOF and circulating cytokine levels across all timepoints. We found that higher expression of many cytokines was associated with increased proportion of myeloid cells. Given our interest in naïve T cell subsets in the manuscript, we also interested to see that IL-6, which was found to have a lower fold change in aged patients in our multivariable analysis (Supplementary Figure 7) had a negative correlation with ThN cells, which were present at lower proportions in aged patients. We present this data for the reviewer here in the response. Size of the dot is related to P value (larger dot = smaller P value).

3. Cytokine Responses to Treatment:

a. Can the authors provide an overview of which cytokines respond to ICI treatment across the entire population and within specific cancer types, irrespective of age?

In our cohort, responders exhibited a decrease in IL-6 and CEGF-A cytokines post-treatment compared to non-responders, and this trend was consistent within the HCC cancer cohort. Since our study primarily focuses on age-related topics and we did not observe a strong relationship between age and treatment response (**Figure 1D**), we presented these results for the reviewer’s information.

Supplementary Figure for Reviewer Response: Box plot of significantly difference cytokines by response for log2 transformed fold change after ICI treatment, in the entire cohort, and within HCC and within RCC cancer groups. P-values were based on Wilcoxon test. P-values in HCC and RCC were with Bonferroni correction.

b. Based on the above, the authors can assess whether the above response patterns are associated with clinical outcomes, such as survival, and examine any dose-response relationship between ICI treatment intensity (i.e. single vs. dual anti ICI therapy) and typical cytokine response.

We thank the reviewer for this suggestion. While our study primarily focuses on age-related topics and we did not observe strong relationships between age and treatment response/survival outcomes (**Figure 1**). Suggested analyses will be explored in future research efforts in our group.

c. Cluster patients based on their cytokine responses and correlate these groups with age, outcome, and irAE.

We appreciate this suggestion and have utilized MDS plots to represent cytokine levels and associate them with both age and ICI response on a per patient basis. We present this for the reviewer below with representations of both baseline (top row) and on treatment (bottom row) timepoints. Patients are separated by age group and then ICI response is denoted (yes vs. no). Briefly, we do not see a clear trend in this low-dimensional representation of the cytokine data. This is consistent with our findings that differences in young and aged patients tend to be in only a select number of cytokines (such as CCL2), and the overall cytokine responses are fairly similar on a global level. We present additional analysis of low-dimensional data in **Figure 6** which is discussed in the response below (**Reviewer #2, Comment #5**).

d. Explore relationships between immune-therapy response parameters (e.g., TMB, MSI) and cytokine baseline and response.

We would also be interested in understanding the impact of aging on ICI responses in clinically significant groups such as those with high TMB or MSI cancers. Unfortunately, we did not have a large enough sample size of young and aged patients with annotated TMB/MSI/MMR data to perform a robust analysis. We will aim to perform these as parts of future studies.

4. Cellular Abundance:

a. In Figure 3C the authors may derive better correlations by regressing cell type abundance against the actual age of patients, rather than calculating differences between age groups. If available, CMV data may be included due to its significant impact on baseline immune variability, particularly in older individuals.

Thank you again for this suggestion. We performed an additional analysis in which we treated patient age as a continuous variable (broken down into decades). Overall, we found similar results as when we used the young and aged groups described in the initially submitted manuscript. Please see Reviewer 1, comment #4 for further details.

Unfortunately, CMV data is not available for the patients in this study, though we agree it is an interesting variable to consider evaluating in future studies.

b. Also here, the authors may investigate the correlation between cellular abundance and tumor type, as observed for cytokines.

We agree that understanding the effect of tumor type on the cellular abundances at baseline and in response to ICI and their association with age is important. To further explore this, we utilized the multivariable analysis models we have described and cited above in this response.

To further investigate this reviewer's suggestion, we also looked at naïve cytotoxic T cell abundances, a major population of interest in our study. Tumor type appeared to not be a major driver of this difference in TcN cells, as aged patients had a lower proportion of TcN cells when considering those with HCC and RCC alone (the two most common tumor types in our cohort) as well as when evaluating all other tumor types with HCC and RCC excluded (**Supplementary Figure 12A-B**), and this is now included in the revised manuscript.

5. Dimensionality and Context: Since all the measurements provided by the manuscript (cytokine levels and cellular abundances) were measured in the same individuals, the authors may provide a low-dimensional representation of patients based on cell subsets abundance and cytokines, showing the positioning of aged versus young individuals, before and after ICI treatment. Can the authors identify a specific axis along which individuals are changing their immune composition over time, thus determining the direction most individuals move toward after treatment in this low-dimensional space. Is this axis associated with outcomes measurements such as irAE and outcome?

We appreciate the suggestion for this interesting analysis. We analyzed the low-dimensional representation of immune profiles by integrating cytokine levels and cell type abundances and performing Principal Component Analysis (PCA) and have included this in the revised manuscript as a newly generated **Figure 6** as well as additional description of these results in the text of the manuscript. Differences in average component scores were evaluated to identify the two principal axes that best facilitated comparisons based on age or response. At baseline, age differences were moderately well-separated by components with relatively high contributions (PC5 and PC7; **Figure 6A**), whereas response differences were not distinctly separated (**Figure 6B**). This result is consistent with our prior analyses because PC5 is directly correlated with naïve T cell proportions while PC7 is inversely correlated with naïve T cell proportions. However, early during treatment, prominent components effectively separated the data by age (PC4; **Figure 6C**) or eventual response (PC2; **Figure 6D**). Here PC4 is driven by a memory/effector T cell signature and is inversely correlated with a naïve T cell signature, and PC2 is representative of a myeloid cell signature. Overall, we find that naïve T cell proportions account for most age-related variance in our data set, and that cellular proportions have an outsized effect compared to cytokine levels, though variances are present across multiple principal components likely due to the underlying heterogeneity of the clinical cohort.

6. Can the authors propose an integrative model using baseline cytokines, age, cellular data, and fold-changes after ICI treatment to predict clinical features? This may be presented a correlation network between cells and cytokines measured in the same individuals, showing how edges change with age and treatment. Highlight how clusters of cells and cytokines predict treatment response.

This is a very exciting idea. We have attempted this using our study, but we believe that building a predictive model will require substantially more patients than currently available in our study. For example, only 25 responders in our current cohort have both cytokine and cellular fold change data, which we believe is not enough for a robust predictive model. This study aimed to broadly profile, identify any strong age-related signals in the cytokine and peripheral immune population data, and generate hypotheses for future analyses. A major ongoing effort in the group is to add patients to this biobanking effort, and in the future, we will pursue this type of work as we increase the number of patients for which we have this information.

7. From a statistical perspective, I suggest ensuring that the comparisons throughout the manuscript are corrected for multiple hypotheses.

We acknowledge the importance of correcting for multiple hypotheses. To control the false discovery rate, we have implemented multiplicity adjustments when assessing the interaction effects between age and treatment

response on cytokine levels and CyTOF outcomes. As other analyses were exploratory and hypothesis-generating, no multiplicity adjustments were made. Clarification was added in the statistical section.

8. Biological Age Stratification: Aged individuals are heterogeneous. I suggest checking the associations identified between cellular abundances and cytokines and outcome to be compared against biological age (determined by baseline immune feature levels) instead of chronological age.

Thank you for your insightful comment. We acknowledge that aged individuals can be quite heterogeneous. However, our study focuses on chronological age rather than biological age, as we do not have available data to stratify our cohort based on biological age. Future research could certainly benefit from examining associations with biological age.

We attempted to perform this investigation but ran up against a few challenges which we believe limited its overall utility in adding to the current study focused on immune responses to ICI in young and aged patients with cancer. Most proposed measures of biological age by baseline immunological features have been established outside the context of a disease state like cancer. For instance, the number of helper T cells and regulatory T cells typically increase with aging and Th1 and Th2 cytokines diminish with aging (<https://doi.org/10.1038/s41392-023-01502-8>), however these changes can also be induced by the presence of cancer (<https://doi.org/10.1038/s41392-024-01868-3>, <https://doi.org/10.1136/jitc-2021-002512>). Therefore, we could not be confident in assigning a patient a biological age independent of their chronological age.

Reviewer #3 (Remarks to the Author): with expertise in human immunology, cancer immunotherapy

Peer review comments for “Age-related divergence of circulating immune responses in patients with solid tumors treated with immunotherapy” by Chester Kao et al., for Nature Communications

Summary

The manuscript “Age-related divergence of circulating immune responses in patients with solid tumors treated with immunotherapy” profiles cancer patient peripheral cytokine levels and immune cell subsets. They analyze the data using the Bioplex 100 platform, Luminex bead-based immunoassays, and CyTOF, and compare differences between younger and aged individuals in a diverse cohort of many cancer types and assess how this impacts response to immune checkpoint inhibitor therapy. Through this method, the authors report that aged patients have differences in immune phenotypes at baseline and on treatment, including reduced levels of naïve T cell with increased immune checkpoint protein expression, as well as an enhanced effector T cell expansion in older responders compared to older non-responders. This study provides an important contribution to the field, however there are several key points that need to be addressed.

Major points

More details are needed on prior oncologic treatment, as the immune profile can be greatly impacted by prior therapies (type, # of lines, etc); some of what is described here may have nothing to do with age, but rather the prior treatment history of the patients evaluated in this study.

We agree that prior treatments received by patients can influence baseline immune features/responses. We have performed additional multivariable analysis which account for patients receiving prior lines of therapy. Overall, these show similar results to our initial analyses, including a reduction in CCL2 production in aged patients after ICI therapy and a diminished pool of circulating naïve T cells in aged patients. We have also added additional information regarding the number of lines of prior therapy received by patients in each age group and show that the average number of prior lines of therapy is not statistically different between young and aged patients (**Table 1** and additional information in the manuscript).

Associations between circulating immune cells (CyTOF) subsets and clinical outcome only include associations with BOR. Did the authors also perform any analyses with PFS or OS as an endpoint? Additional key features may be uncovered that do not associate with BOR but do associate with longer term outcomes of patients.

Thank you for your suggestion. While our study primarily focuses on age-related topics and we did not observe strong relationships between age and treatment response/survival outcomes (**Figure 1**), analyses in this area will be conducted for further studies. We favored using BOR as we could use RECIST criteria to clearly define response and make it a discrete variable. Given the heterogeneity of our cohort that has multiple tumor types, using PFS or OS was less attractive as different tumor types have vastly different expected PFS and OS periods with these treatments (for instance HCC vs. melanoma).

Although there were no age associated differences at baseline in circulating analytes (cytokines), were there any associations between circulating analytes and clinical outcomes (either in the combined age cohort, the old cohort, the young cohort, or in patients with HCC or RCC (your two biggest cancer type cohorts)? In addition, since there were cancer type associated differences in cytokines shown in Extended Data Figure 1, were there any age-related differences in cytokines when broken down into HCC or RCC (your two biggest cancer type cohorts)?

We thank the reviewer for this question. To address it, we have included a new multivariable analysis of cytokine levels exploring age-related differences in ICI response (**Supplementary Figure 9**). We note interesting new trends that were uncovered by this analysis in the manuscript. For instance, aged responders tended to have decreased CCL2 at baseline compared to aged non-responders, which is the opposite of what we observe in young responders vs. non responders. Further, while aged patients across the cohort have lower fold change in CCL2 post ICI therapy compared to young patient, we find that aged responders actually have an increased CCL2 fold change compared to non-responders.

In this response, we have included a cytokine analysis of cytokines in young and aged patients with HCC and another for patients with RCC. We find similar trends to the analysis performed across the entire cohort, though fewer significant differences, which may be attributed to reduced statistical power. Therefore, we believe the analysis of the entire cohort represents the best overall picture of age-related differences in these circulating analytes.

When were post treatment bloods collected and analyzed? In the results, you state that blood was collected approximately 1-6 months after initiation of a checkpoint inhibitor therapy (line 105). In the methods, you state that peripheral blood samples were collected at month 1, 2, 4, 6, and 12 (line 393). In the figures, the timepoint is just referred to as "on-treatment". Which timepoint(s) was(were) used in the analysis? A 6th month window is likely too wide of a window post treatment and should be narrowed, even if the n for paired analyses pre and post treatment is much smaller. Many changes in circulating analytes, for example that are seen a month after initiating immune checkpoint blockade are no longer seen after multiple cycles of treatment, which may impact the results of the current study. If the window of 6 month is kept, you really need to add this as a variable in the heatmap of anything where you are looking at post or on-treatment timepoints.

We agree that this is important to clarify, and we have updated the manuscript to be more specific regarding the timepoints used for analyses in this study. 83.2% (n=79) of patients had on treatment samples collected within 2 months of starting ICI therapy, with only 3.2% (n=3) of patients having samples collected in months 3-5 post-initiation of ICI therapy, which we believe will represent uniformity in overall temporal dynamics across these specimens.

Do any changes in circulating analytes after therapy associate with clinical outcomes (either in the combined age cohort, the old cohort, the young cohort, or in patients with HCC or RCC (your two biggest cancer type cohorts)?

We have provided additional analysis to directly address this question with a multivariable model that looks at response vs. non-response in young and aged patients (**Supplementary Figure 7** and **Supplementary Figure 14**). We identified unique changes in cytokines including CCL2 that associate with response that we further discuss in the revised manuscript. We also present cytokine analysis by age in patients with HCC and RCC as separate analyses (**Supplementary Figure 5** and **Supplementary Figure 7**).

It should be noted that many of the immune cell subset differences described between young and old patients in this study are not novel (e.g. naïve T cell subsets, NK cells).

We appreciate that these are not novel immune cell subsets we have described, and we will have more clearly stated this in the revised manuscript. We believe that by providing information on these established cell subsets in the context of immunotherapy response and aging we can aid others in designing studies (including clinical trials) that can monitor or use this information to understand the impacts of aging on anti-tumor immune responses.

Do the detection antibodies utilized to detect PD-1, PD-L1, CTLA4 with the CyTof assay compete with the checkpoint therapies some of these patients have received previously, or are treated with in the current study? This should be considered in interpreting data from any patients post treatment with checkpoint blockade.

The CyTOF antibodies do not compete with the checkpoint therapies received by the patients in the study with the exception of the antibody for PD-1 (clone EH12.2H7). In light of this, we did not present any conclusions about PD-1 expression across time points (for instance saying that aged patients have greater or lesser treatment-related changes in PD-1 expression) and rather strictly compared PD-1 expression either before or after treatment. Therefore, we believe any potential competition applies evenly across patients. We also are then able to use this fact as a positive control in our workflow that shows that across all patients PD-1 expression is decreased with ICI treatment.

Line 275. Did you perform any comparisons of responders vs non-responders for the analysis of immune checkpoints? Since aged patients have increased expression of immune checkpoints in their naïve T cells, how does this specifically relate to response to therapy, for example?

To evaluate differences in immune checkpoint expression in naïve T cells and if it differs by patient response, we compared baseline expression of several immune checkpoints in the naïve T cell clusters (such as TcN) of young and aged patients. In aged patients, responders had lower PD-1 levels on TcN cells compared to non-responders ($P = 0.047$) and trends toward lower TIGIT and CTLA-4 expression which were non-significant. In young patients, we also observed lower PD-1 on TcN in responders ($P = 0.037$). We have included these findings in a new supplemental figure in the manuscript (**Supplementary Figure 19**). We did not find any statistically significant differences in ThN cells nor in TcEFF cells in either young or aged responders vs. non-responders.

Was an age cutoff other than 65 considered for this work?

Please see our response to Reviewer 1, item #4 for a discussion of the age cutoff chosen for this work.

Many of the figures are not legible when printed – the text is far too small and blurry which makes the study extremely difficult to interpret/review. The following figures especially need to be enlarged/improved: Figure 2B (heatmap), Figure 3C and D (graphs and text too small), Figure 4 – text too small, all graphs and heatmaps in Figure 5 (can't read anything).

We apologize for the difficulty in reading these figures caused by small fonts/graphics. We have re-arranged the figures to allow us to increase the size and legibility of figure panels for this revised manuscript submission.

Minor points

Last line of the introduction – starting on line 94 is an incomplete sentence – needs to be re-worded.

We have revised this sentence to correct this error and apologize for our oversight.

More details on PD-L1 classification is needed.

For PD-L1 classification, we identified patients with a commercially available molecular profiling assay (including but not limited to Tempus, Caris, and Foundation Medicine) as standard of care. PD-L1 classification (none, low, high) was based on the assay utilized and directly reported by these commercial entities, and thus as it would be interpreted by the ordering physician in a standard of care treatment paradigm. If a patient had multiple types

of PD-L1 assessments such as tumor proportion score (TPS) and combined positive score (CPS) available, then the PD-L1 value used for classification was the type of PD-L1 assessment utilized for that specific tumor type per national guidelines (such as the NCCN guidelines).

Line 378. Suggest ending with conclusions instead of limitations.

We have reordered points of the discussion to end with conclusions instead of listing limitations.

Running title should be changed to Cellular responses; there are no Molecular responses being evaluated in the current study.

Thank you for this suggestion, we have incorporated this into our revised manuscript.

Supplemental tables need to be combined so all columns are on a single page – can't make sense of them as they are now – spread out on multiple pages

We believe this may be an artifact of the manuscript processing that occurs as part of the uploading process. Supplemental tables that we have uploaded all fit on a single 8.5"x11" Word Document with standard margins aside from Supplemental Table 4 and Supplemental Table 5 which contain data that is best visualized using spreadsheet software such as Microsoft Excel. We will work with the editorial team to address any issues if they arise again during review in this resubmission process.

When discussing post treatment analytes, please be clear whether absolute levels post treatment or changes compared to baseline are being evaluated.

Thank you for this reminder, we have added more specificity throughout the manuscript regarding these comparisons that occur after treatment as either on treatment (representations absolute levels in the on treatment timepoint specimen) or fold change (a ratio of absolute levels from on treatment compared to baseline specimens).

Response to Reviewers

We thank the reviewers for their time and effort spent in evaluating our revised manuscript. It is clear the reviewers thoroughly and thoughtfully evaluated the manuscript and our responses, and it has helped us improve this study. Here we present responses to Reviewer #3's additional comments and suggestions with Reviewer #3's comments in **RED** and our responses in **BLUE**.

Reviewer #3 (Remarks to the Author): with expertise in human immunology, cancer immunotherapy

Peer review comments for "Age-related divergence of circulating immune responses in patients with solid tumors treated with immunotherapy" by Chester Kao et al., for Nature Communications

Summary

The manuscript "Age-related divergence of circulating immune responses in patients with solid tumors treated with immunotherapy" profiles cancer patient peripheral cytokine levels and immune cell subsets. They analyze the data using the Bioplex 100 platform, Luminex bead-based immunoassays, and CyTOF, and compare differences between younger and aged individuals in a diverse cohort of many cancer types and assess how this impacts response to immune checkpoint inhibitor therapy. Through this method, the authors report that aged patients have differences in immune phenotypes at baseline and on treatment, including reduced levels of naïve T cell with increased immune checkpoint protein expression, as well as an enhanced effector T cell expansion in older responders compared to older non-responders. This study provides an important contribution to the field, however there are several key points that need to be addressed.

Major points

More details are needed on prior oncologic treatment, as the immune profile can be greatly impacted by prior therapies (type, # of lines, etc); some of what is described here may have nothing to do with age, but rather the prior treatment history of the patients evaluated in this study.

We agree that prior treatments received by patients can influence baseline immune features/responses. We have performed additional multivariable analysis which account for patients receiving prior lines of therapy. Overall, these show similar results to our initial analyses, including a reduction in CCL2 production in aged patients after ICI therapy and a diminished pool of circulating naïve T cells in aged patients. We have also added additional information regarding the number of lines of prior therapy received by patients in each age group and show that the average number of prior lines of therapy is not statistically different between young and aged patients (**Table 1** and additional information in the manuscript).

Thank you – I have no further comments.

Associations between circulating immune cells (CyTOF) subsets and clinical outcome only include associations with BOR. Did the authors also perform any analyses with PFS or OS as an endpoint? Additional key features may be uncovered that do not associate with BOR but do associate with longer term outcomes of patients.

Thank you for your suggestion. While our study primarily focuses on age-related topics and we did not observe strong relationships between age and treatment response/survival outcomes (**Figure 1**), analyses in this area will be conducted for further studies. We favored using BOR as we could use RECIST criteria to clearly define response and make it a discrete variable. Given the heterogeneity of our cohort that has multiple tumor types, using PFS or OS was less attractive as different tumor types have vastly different expected PFS and OS periods with these treatments (for instance HCC vs. melanoma).

Given that more than one reviewer brought this up, I suggest to expand upon this in the discussion; readers of this paper may likely have the same thought. You start to address this in lines 537-539, "The lack of a significant association between patient age and PFS and OS may be attributed to the short follow-up period of the study coupled with the variety of tumor types represented in the cohort, each with differing expected clinical courses", but I would suggest to specify that you restricted your detailed analyses correlating immune responses with patient response to therapy using BOR/RECIST, as PFS/OS would be unsuitable given the high degree of heterogeneity of your patient population.

Thank you for this valuable suggestion. We agree that additional specificity will help readers evaluate our study. To that end, we have incorporated the following statement into the revised manuscript: "In addition, we limited our detailed analyses correlating immune responses with therapy responses in patients using BOR/RECIST as PFS/OS would likely be unsuitable given the high degree of heterogeneity (multiple tumor types with variable disease courses) in our cohort."

Although there were no age associated differences at baseline in circulating analytes (cytokines), were there any associations between circulating analytes and clinical outcomes (either in the combined age cohort, the old cohort, the young cohort, or in patients with HCC or RCC (your two biggest cancer type cohorts)? In addition, since there were cancer type associated differences in cytokines shown in Extended Data Figure 1, were there any age-related differences in cytokines when broken down into HCC or RCC (your two biggest cancer type cohorts)?

We thank the reviewer for this question. To address it, we have included a new multivariable analysis of cytokine levels exploring age-related differences in ICI response (**Supplementary Figure 9**). We note interesting new trends that were uncovered by this analysis in the manuscript. For instance, aged responders tended to have decreased CCL2 at baseline compared to aged non-responders, which is the opposite of what we observe in young responders vs. non responders.

Further, while aged patients across the cohort have lower fold change in CCL2 post ICI therapy compared to young patients, we find that aged responders actually have an increased CCL2 fold change compared to non-responders. In this response, we have included a cytokine analysis of cytokines in young and aged patients with HCC and another for patients with RCC. We find similar trends to the analysis performed across the entire cohort, though fewer significant differences, which may be attributed to reduced statistical power. Therefore, we believe the analysis of the entire cohort represents the best overall picture of age-related differences in these circulating analytes.

Thank you for adding this analysis. For Supplementary Figure 9, the scale of the coefficient plot needs to be corrected.

We thank the reviewer for catching this mistake in our figure preparation. The x axis scales for both plots were indeed incorrect in the previously submitted manuscript, and we have corrected Supplementary Figure 9 to fix this issue.

When were post treatment bloods collected and analyzed? In the results, you state that blood was collected approximately 1-6 months after initiation of a checkpoint inhibitor therapy (line 105). In the methods, you state that peripheral blood samples were collected at month 1, 2, 4, 6, and 12 (line 393). In the figures, the timepoint is just referred to as "on-treatment". Which timepoint(s) was(were) used in the analysis? A 6th month window is likely too wide of a window post treatment and should be narrowed, even if the n for paired analyses pre and post treatment is much smaller.

Many changes in circulating analytes, for example that are seen a month after initiating immune checkpoint blockade are no longer seen after multiple cycles of treatment, which may impact the results of the current study. If the window of 6 month is kept, you really need to add this as a variable in the heatmap of anything where you are looking at post or on-treatment timepoints.

We agree that this is important to clarify, and we have updated the manuscript to be more specific regarding the timepoints used for analyses in this study. 83.2% (n=79) of patients had on treatment samples collected within 2 months of starting ICI therapy, with only 3.2% (n=3) of patients having samples collected in months 3-5 post-initiation of ICI therapy, which we believe will represent uniformity in overall temporal dynamics across these specimens.

Thank you for adding this additional clarifying information. However, you indicate that there are 95 patients with on-treatment samples evaluated. These numbers 79+3 do not add up to 95. When were post-treatment samples evaluated from the remaining 13 patients (out of 95) who had on-treatment samples assessed?

We apologize for the confusion and agree that the way this was written is not specific enough. The additional 13 patients mentioned by the reviewer had samples collected after the end of month 2 but before the end of month 3 post-ICI therapy. We have changed the manuscript to say: "83.2% (n=79) of patients had on treatment samples collected within 2 months of starting ICI therapy, 13.7% (n=13) collected after 2 months but within 3 months, and only 3.2% (n=3) of patients having samples collected 3-5 months post-initiation of ICI therapy."

Do any changes in circulating analytes after therapy associate with clinical outcomes (either in the combined age cohort, the old cohort, the young cohort, or in patients with HCC or RCC (your two biggest cancer type cohorts)?

We have provided additional analysis to directly address this question with a multivariable model that looks at response vs. non-response in young and aged patients (**Supplementary Figure 7** and **Supplementary Figure 14**). We identified unique changes in cytokines including CCL2 that associate with response that we further discuss in the revised manuscript. We also present cytokine analysis by age in patients with HCC and RCC as separate analyses (**Supplementary Figure 5** and **Supplementary Figure 7**).

Thank you for this addition; I have no further comments on this.

It should be noted that many of the immune cell subset differences described between young and old patients in this study

are not novel (e.g. naïve T cell subsets, NK cells).

We appreciate that these are not novel immune cell subsets we have described, and we will have more clearly stated this in the revised manuscript. We believe that by providing information on these established cell subsets in the context of immunotherapy response and aging we can aid others in designing studies (including clinical trials) that can monitor or use this information to understand the impacts of aging on anti-tumor immune responses.

It is not clear where this is updated in the manuscript. At lines 551-552, you indicate that your study is “among the first to prospectively and deeply phenotype the circulating immune cell populations and cytokine responses to ICI treatment, and to tie unique phenotypes to patient age”, but you do not provide citations of others who have done so and their findings. Perhaps this would be a good section of the discussion to expand upon how your findings agree/don’t agree with what has been previously reported.

We thank the reviewer for this suggestion and agree further discussion of our specific results in context of other studies is a valuable addition. Therefore, we have expanded our discussion to more directly address these points. In summary, we are not aware of another prospective study that deeply profiles circulating immune cells, analyzes circulating cytokine levels and then correlates these findings with patient age and ICI response across tumor types. Mostly, other studies (References 66-70 in the revised manuscript) describe one of these parameters (such as circulating immune cells alone or cytokines alone), typically without the depth of phenotyping available to use with our CyTOF panel, and often not making clear distinctions between young and older patients. Further, many prior studies have focused globally on “senescence” in T cell populations, while our study highlights differences in naïve T cell populations between younger and older patients. Thank you again for the opportunity to improve the manuscript.

Do the detection antibodies utilized to detect PD-1, PD-L1, CTLA4 with the CyToF assay compete with the checkpoint therapies some of these patients have received previously, or are treated with in the current study? This should be considered in interpreting data from any patients post treatment with checkpoint blockade.

The CyTOF antibodies do not compete with the checkpoint therapies received by the patients in the study with the exception of the antibody for PD-1 (clone EH12.2H7). In light of this, we did not present any conclusions about PD-1 expression across time points (for instance saying that aged patients have greater or lesser treatment-related changes in PD-1 expression) and rather strictly compared PD-1 expression either before or after treatment. Therefore, we believe any potential competition applies evenly across patients. We also are then able to use this fact as a positive control in our workflow that shows that across all patients PD-1 expression is decreased with ICI treatment.

I don’t think that PD-1 should be included in any post treatment analysis where patients were treated with anti-PD-1 therapies if the antibody competes with the therapy. For example, in figure 5, most of these patients would have received anti-PD-1 therapy and a small portion would have received anti-PD-L1. If there is competitive binding, how can you know it is even across patients, given the variation in checkpoint treatment among your cohort? I suggest to indicate that this antibody competes in the methods section and that PD-1 expression was thus not considered for any post-treatment analyses.

We really appreciate the thoughtful comment from the reviewer regarding the anti-PD-1 antibody used in this study. After considering these points we are also inclined to agree with the reviewer’s suggestion. Therefore, we have removed data comparing PD-1 expression in post-treatment analyses (Such as in Figure 5 and Supplemental Figure 18. We have placed a black X over the PD1 portions of the On Treatments to clearly indicate these were not considered in our analyses. We have also included that information in the figure legends and added the following statement to the methods The PD-1 antibody used in the CyTOF analysis

competes with anti-PD-1 therapies received by patients in this study. Therefore, we did not consider PD-1 expression for any on treatment timepoint analyses.

Line 275. Did you perform any comparisons of responders vs non-responders for the analysis of immune checkpoints? Since aged patients have increased expression of immune checkpoints in their naïve T cells, how does this specifically relate to response to therapy, for example?

To evaluate differences in immune checkpoint expression in naïve T cells and if it differs by patient response, we compared baseline expression of several immune checkpoints in the naïve T cell clusters (such as TcN) of young and aged patients. In aged patients, responders had lower PD-1 levels on TcN cells compared to non-responders ($P = 0.047$) and trends toward lower TIGIT and CTLA-4 expression which were non-significant. In young patients, we also observed lower PD-1 on TcN in responders ($P = 0.037$). We have included these findings in a new supplemental figure in the manuscript (**Supplementary Figure 19**). We did not find any statistically significant differences in ThN cells nor in TcEFF cells in either young or aged responders vs. non-responders.

Thank you for this addition; I have no further comments on this.

Was an age cutoff other than 65 considered for this work?

Please see our response to Reviewer 1, item #4 for a discussion of the age cutoff chosen for this work.

No further comments.

Many of the figures are not legible when printed – the text is far too small and blurry which makes the study extremely difficult to interpret/review. The following figures especially need to be enlarged/improved: Figure 2B (heatmap), Figure 3C and D (graphs and text too small), Figure 4 – text too small, all graphs and heatmaps in Figure 5 (can't read anything).

We apologize for the difficulty in reading these figures caused by small fonts/graphics. We have re-arranged the figures to allow us to increase the size and legibility of figure panels for this revised manuscript submission.

The appearance of the figures is much improved.

Minor points

Last line of the introduction – starting on line 94 is an incomplete sentence – needs to be re-worded.

We have revised this sentence to correct this error and apologize for our oversight.

No further comments.

More details on PD-L1 classification is needed.

For PD-L1 classification, we identified patients with a commercially available molecular profiling assay (including but not limited to Tempus, Caris, and Foundation Medicine) as standard of care. PD-L1 classification (none, low, high) was based on the assay utilized and directly reported by these commercial entities, and thus as it would be interpreted by the ordering physician in a standard of care treatment paradigm. If a patient had multiple types of PD-L1 assessments such as tumor proportion score (TPS) and combined positive score (CPS) available, then the PD-L1 value used for classification was the type of PD-L1 assessment utilized for that specific tumor type per national guidelines (such as the NCCN guidelines).

No further comments.

Line 378. Suggest ending with conclusions instead of limitations.

We have reordered points of the discussion to end with conclusions instead of listing limitations.

No further comments.

Running title should be changed to Cellular responses; there are no Molecular responses being evaluated in the current study.

Thank you for this suggestion, we have incorporated this into our revised manuscript.

No further comments.

Supplemental tables need to be combined so all columns are on a single page – can't make sense of them as they are now spread out on multiple pages

We believe this may be an artifact of the manuscript processing that occurs as part of the uploading process. Supplemental tables that we have uploaded all fit on a single 8.5"x11" Word Document with standard margins aside from Supplemental Table 4 and Supplemental Table 5 which contain data that is best visualized using spreadsheet software such as Microsoft Excel. We will work with the editorial team to address any issues if they arise again during review in this resubmission process.

No further comments.

When discussing post treatment analytes, please be clear whether absolute levels post treatment or changes compared to baseline are being evaluated.

Thank you for this reminder, we have added more specificity throughout the manuscript regarding these comparisons that occur after treatment as either on treatment (representations absolute levels in the on treatment timepoint specimen) or fold change (a ratio of absolute levels from on treatment compared to baseline specimens).

No further comments.

Additional Minor comment:

The word "responders" is missing in line 370: "Forty-six of these patients were >65 years old of which 17 were classified as and 29 as non-responders (ORR = 37.0%)"

We thank the reviewer for the careful reading of our manuscript and apologize for this error. We have corrected it by adding "responders" as suggested.